# IMAGINE HOW TO CHANGE: EXPLICIT PROCEDURE MODELING FOR CHANGE CAPTIONING

**Jiayang Sun**[*1]  **Zixin Guo**[*2]  **Min Cao**[†1]  **Guibo Zhu**[†345]  **Jorma Laaksonen**[2]

[1]School of Computer Science and Technology, Soochow University, Jiangsu, China
[2]Department of Computer Science, Aalto University, Espoo, Finland
[3]Foundation Model Research Center, Institute of Automation, Chinese Academy of Sciences, Beijing, China
[4]School of Artificial Intelligence, University of Chinese Academy of Sciences, Beijing, China
[5]Wuhan AI Research

jysun02@stu.suda.edu.cn   zixin.guo@aalto.fi
mcao@suda.edu.cn   gbzhu@nlpr.ia.ac.cn

## ABSTRACT

Change captioning generates descriptions that explicitly describe the differences between two visually similar images. Existing methods operate on static image pairs, thus ignoring the rich temporal dynamics of the change procedure, which is the key to understand not only what has changed but also how it occurs. We introduce ProCap, a novel framework that reformulates change modeling from static image comparison to dynamic procedure modeling. ProCap features a two-stage design: The first stage trains a procedure encoder to learn the change procedure from a sparse set of keyframes. These keyframes are obtained by automatically generating intermediate frames to make the implicit procedural dynamics explicit and then sampling them to mitigate redundancy. Then the encoder learns to capture the latent dynamics of these keyframes via a caption-conditioned, masked reconstruction task. The second stage integrates this trained encoder within an encoder-decoder model for captioning. Instead of relying on explicit frames from the previous stage—a process incurring computational overhead and sensitivity to visual noise—we introduce learnable procedure queries to prompt the encoder for inferring the latent procedure representation, which the decoder then translates into text. The entire model is then trained end-to-end with a captioning loss, ensuring the encoder's output is both temporally coherent and captioning-aligned. Experiments on three datasets demonstrate the effectiveness of ProCap. Code and pre-trained models are available at https://github.com/BlueberryOreo/ProCap.

## 1 INTRODUCTION

Change captioning aims to generate textual descriptions that emphasize differences between two similar images. It has attracted growing interest due to its wide applications, like monitoring temporal changes in remote sensing (Chouaf et al., 2021), supporting medical diagnosis by leveraging comparisons between abnormal and normal medical images (Bian et al., 2025), supporting urban planning via intelligent surveillance (Sun et al., 2024), and improving industrial quality control (Xie et al., 2024). Despite its promise, the task remains challenging due to (1) subtle appearance changes often being obscured by variations in viewpoint, illumination, or background clutter, and (2) the difficulty of transforming fine-grained visual differences into coherent, accurate language descriptions.

To address them, existing methods follow an encoder-decoder framework, where the encoder captures visual differences and the decoder generates descriptive captions. Early works (Park et al., 2019; Shi et al., 2020) model pixel-level differences via patch features, while later works (Qiu et al., 2021; Yao et al., 2022; Tu et al., 2023c) introduce intricate difference extractors with alignment

---

[*]Both authors contributed equally to this research.
[†]Corresponding authors.

mechanisms to better localize change regions. More recently, the field has seen a shift towards integrating Large Language Models (LLMs) as decoders (Yang et al., 2023; Hu et al., 2024; Zhang et al., 2024), leading to substantial gains in caption quality. Furthermore, recent advancements in applying reinforcement learning to bolster LLM reasoning (Peng et al., 2025; Wu et al., 2025b) present a promising avenue for further enhancing change captioning. Although promising, these methods typically focus on static image pairs, neglecting dynamic context and temporal cues critical for robust change perception. In practice, the transition between images often involves intermediate frames that capture rich spatio-temporal dynamics, explicitly revealing appearance and motion changes only implicitly encoded in the static pair (see Figure 1). Explicitly modeling this transition process thus offers a more principled basis for change understanding and captioning.

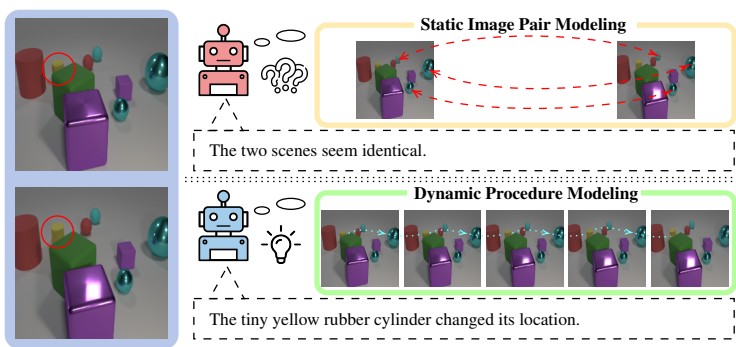

Figure 1: Comparison between static image pair modeling and our proposed dynamic procedure modeling. Dynamic procedures offer temporal cues: the yellow cylinder, initially partly obscured by the green cube, changes its location.

In this work, we make the first attempt beyond change captioning on static image pairs by formulating a *procedure-modeling-then-captioning* paradigm. We explicitly model the dynamic change procedure between static image pairs and perform captioning on the modeled spatio-temporal change procedure. We present ProCap, an innovative two-stage framework: (1) explicit procedure modeling, which captures latent spatio-temporal dynamics between image pairs, and (2) implicit procedure captioning, which generates rich descriptions by leveraging learnable queries to implicitly reason over the modeled change procedure.

**Explicit procedure modeling.** Our framework models the underlying change procedure via three key components. *Procedure Generation Module*: This component synthesizes intermediate frames to transform the implicit transformation between input images into an explicit and observable temporal sequence. However, the generated sequence tends to be dense and temporally redundant, often containing low-information content that incurs unnecessary computational overhead. *Confidence-Based Frame Sampling Module*: To address this, we introduce a confidence-aware sampling module to distill the sequence into a sparse set of informative keyframes. Each frame is assigned a confidence score based on temporal and semantic importance. By retaining only the highest-scoring frames, our module focuses learning on pivotal transition moments, thereby improving efficiency and representational quality during training. *Procedure Modeling Module*: Finally, we employ a procedure encoder to learn a compact latent representation of the sampled keyframe sequence. We cast this as a caption-conditioned masked frame reconstruction task, where a multi-granularity masking strategy—ranging from local patches to entire frames—is introduced. This encourages the model to capture aligned spatio-temporal dynamics across multiple scales, while mitigating overfitting to superficial visual cues and enhancing generalization in procedural understanding.

**Implicit procedure captioning.** The key challenge for captioning lies in leveraging the procedural knowledge learned by the procedure encoder for efficient and effective text generation. A naive approach—generating and encoding intermediate frames at inference—incurs high computational cost and introduces sensitivity to synthesis noise. To address them, we present the implicit procedure captioning, inserting a set of learnable procedure queries between static image pairs, acting as "slots" to replace explicit intermediate frames. By leveraging the understanding of spatio-temporal dynamics learned during the first stage, the procedure encoder is prompted to infer the latent change

procedure implicitly encoded within the image pair. The resulting procedural representation is decoded into a textual description, enabling end-to-end optimization via a captioning loss. This yields a temporally coherent, task-aligned representation without costly frame synthesis at inference.

Our contributions are summarized: (1) We introduce ProCap, a two-stage framework reformulating change captioning from static comparison to dynamic procedure modeling, directly addressing limitation of prior works: their reliance on static image pairs, overlooking rich temporal dynamics. (2) We propose explicit procedure modeling, where a procedure encoder is trained on sampled keyframes from a synthesized explicit procedure, with a caption-conditioned masked reconstruction task to capture change dynamics. (3) We develop implicit procedure captioning, introducing learnable queries to enable the encoder to model the procedure implicitly, bypassing the costly and noise-prone frame synthesis at inference for efficient and effective captioning.

## 2   RELATED WORK

Existing change captioning methods primarily operate on static image pairs, treating the task as a spatial comparison problem. Pioneering works by Jhamtani & Berg-Kirkpatrick (2018) and Park et al. (2019) establish a foundational encoder-decoder framework. Subsequent efforts enhance this static comparison with two main paradigms: (1) designing intricate change encoders for fine-grained localization and robustness to distractors like viewpoint or illumination changes (Kim et al., 2021; Tu et al., 2023a; Yue et al., 2023; Tu et al., 2024a; Li et al., 2025; Hu et al., 2025; Zhong et al., 2025); and (2) adopting advanced training strategies, such as auxiliary retrieval tasks (Hosseinzadeh & Wang, 2021) or multi-stage alignment (Guo et al., 2022; Yao et al., 2022; Rahmanzadehgervi et al., 2025), to guide the learning process. Beyond model design, recent vision-language studies (Menon & Vondrick, 2022; Pratt et al., 2023; Guo et al., 2023) have increasingly explored prompt-driven methodologies to harness large-scale data. Inspired by these advancements, recent studies like Liu et al. (2025) and Di et al. (2025) have focused on constructing large-scale, high-quality datasets to further advance the field. Furthermore, to mitigate hallucinations arising from large-scale datasets, Guo et al. (2025) explores a noise-robust pre-training framework for change captioning. Although promising, these methods infer changes directly from "before" and "after" images, ignoring the underlying continuous and dynamic transition process. In contrast, we propose to explicitly model the change procedure, shifting the paradigm from spatial comparison to spatio-temporal procedure modeling. We argue that the intermediate sequence contains rich temporal dynamics critical for robust change understanding—information inherently missing in static pairs. While recent advances in video understanding have focused on improving temporal grounding in LLMs through explicit frame identifiers (Wu et al., 2025a), the application of such dynamic modeling in change captioning remains underexplored. The most closely related work is Zhu et al. (2025), which implicitly models temporal dynamics in remote sensing using domain-specific change maps. Our approach differs fundamentally: (1) we explicitly generate and model intermediate transitions to reason about how changes unfold, enhancing dynamic representation; and (2) we eliminate reliance on domain-specific supervision, enabling generalization to complex, unconstrained natural scenes. Additional related work, particularly on frame interpolation, is included in Appendix B.

## 3   METHODOLOGY

Relying solely on two static images, existing methods neglect the rich spatio-temporal procedure that connects an image pair. Our key insight is that such procedure is crucial for understanding not only what has changed but also how it occurs, thereby improving the change dynamics modeling. Given an image pair $(I_{\text{bef}}, I_{\text{aft}})$ containing objects $O = \{o_1, o_2, \ldots, o_n\}$, each object $o_i$ is represented by three continuous attributes $(p_i, a_i, w_i)$ corresponding to position, appearance, and existence, respectively. A valid change procedure with respect to a change caption $T$ is formalized as a mapping $\gamma_T : [0, 1] \to \mathcal{I}$, where $\mathcal{I}$ denotes the space of all possible images, satisfying: (1) boundary conditions $\gamma_T(0) = I_{\text{bef}}$ and $\gamma_T(1) = I_{\text{aft}}$; (2) continuous evolution of each object's attributes $(p_i, a_i, w_i)$ over time $t$, such that $\gamma_T(t) = \{(p_i(t), a_i(t), w_i(t))\}_{i=1}^n$; (3) consistency with the semantic constraints imposed by caption $T$; and (4) invariance of unchanged objects throughout the process. Our objective is to derive an informative sequence $\mathcal{P} \subset \mathcal{I}$ that approximates $\gamma_T$. To address this, we introduce procedure modeling for captioning (ProCap), illustrated in Figure 2. Our proposed ProCap formulates change captioning as a two-stage learning process: (1) explicit proce-

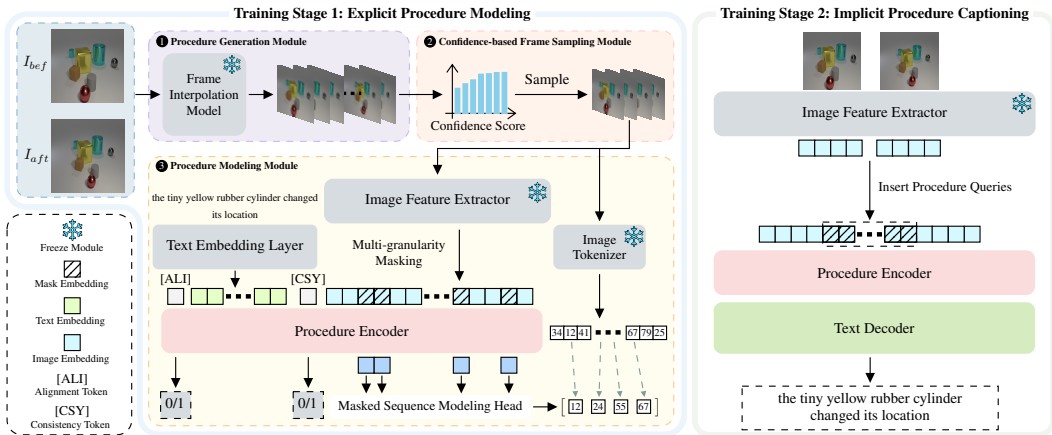

Figure 2: Our two-stage ProCap framework. In the first stage, Explicit Procedure Modeling, a procedure encoder learns change dynamics from keyframes sampled from the generated explicit procedure frames. In the second stage, Implicit Procedure Captioning, learnable procedure queries, instead of explicit frames, prompt the encoder to infer an implicit representation for captioning.

dure modeling stage learning to capture the latent dynamics of the change procedure, and (2) implicit procedure captioning stage learning to generate descriptions based on the modeled procedure.

### 3.1 EXPLICIT PROCEDURE MODELING

This stage incorporates three key components: a procedure generation module that produces continuous frames between given static image pairs, a confidence-based frame sampling module that identifies and selects keyframes from the produced continuous frames, and a procedure modeling module that models the latent change dynamics in these keyframes.

#### 3.1.1 PROCEDURE GENERATION MODULE

The first step is to make the change procedure explicit. To achieve this, we employ a pre-trained, off-the-shelf Frame Interpolation (FI) model (Lu et al., 2022) to synthesize the procedure. Given an image pair $(I_{\text{bef}}, I_{\text{aft}})$, the FI model first uses a CNN to predict bidirectional optical flows $\{\boldsymbol{O}_{t\to\text{bef}}, \boldsymbol{O}_{t\to\text{aft}}\}$, which are applied to the images and their features to generate warped image pairs $(\tilde{I}_{\text{bef}}, \tilde{I}_{\text{aft}})$ and warped feature pairs $(\tilde{\boldsymbol{F}}_{\text{bef}}, \tilde{\boldsymbol{F}}_{\text{aft}})$. These warped pairs, along with the original images, are then fed into a Transformer that produces a soft mask $\boldsymbol{H}$ and an image residual $\Delta I_t$. The intermediate frame $I_t$ is synthesized as $I_t = \boldsymbol{H} \odot \tilde{I}_{\text{bef}} + (1 - \boldsymbol{H}) \odot \tilde{I}_{\text{aft}} + \Delta I_t$, where $\odot$ denotes the Hadamard product. Typically, $I_t$ represents an intermediate state within the overall change procedure. To construct a sequence of $l$ pseudo-frames, we recursively apply the FI model, yielding an explicit procedure:

$$\mathcal{P}^{\text{FI}} = \text{FI}(I_{\text{bef}}, I_{\text{aft}}) = \{I_1, I_2, \ldots, I_l\}. \tag{1}$$

However, the generated dense sequence $\mathcal{P}^{\text{FI}}$ is not optimal for direct modeling (Appendix M.2, Figure 9). The primary challenge is inherent temporal redundancy. Owing to the recursive nature of the synthesis process, redundancy in a single intermediate frame—particularly when it closely resembles the input images—propagates to adjacent regions of the sequence, thereby providing minimal novel information about the change. Modeling this entire sequence is not only computationally inefficient but also risks diluting the critical moments of the change with trivial, redundant frames. Therefore, distilling the sequence into a sparse set of keyframes that are relatively more informative about the change dynamics is critical for efficient and effective procedure modeling.

#### 3.1.2 CONFIDENCE-BASED FRAME SAMPLING MODULE

To achieve this, we introduce a confidence-based frame sampling module. It identifies and selects the keyframes using a "score-then-sample" strategy. Specifically, each frame in $\mathcal{P}^{\text{FI}}$ is assigned a confidence score quantifying its informativeness, which then guides the sampling process.

**Score.** To quantify a frame's informativeness, we formalize the intuition that the more critical frames are those that represent the semantic midpoint of the change—the point where a frame is semantically equidistant from the initial ($I_{\text{bef}}$) and final ($I_{\text{aft}}$) states. Conversely, frames that are highly similar to either endpoint are information-redundant. Given the image pair ($I_{\text{bef}}, I_{\text{aft}}$) and the generated set $\mathcal{P}^{\text{FI}}$, we compute the confidence score vector $\boldsymbol{w}$ as:

$$\boldsymbol{w} = 1 - \sigma(\,[\,s(I_{\text{bef}}, \mathcal{P}^{\text{FI}}) - s(I_{\text{aft}}, \mathcal{P}^{\text{FI}})]^2\,), \tag{2}$$

where $\sigma(\cdot)$ is the softmax function, and $s(\cdot, \cdot)$ is a semantic similarity function. The squared difference term ensures that frames are penalized regardless of which endpoint they are closer to. This score $\boldsymbol{w}$ assigns high values to frames that are semantically equidistant from the start and end images, effectively identifying the "peak" of the change.

We explore two strategies to compute the similarity $s(\cdot, \cdot)$, leveraging different sources of information: (1) visual-only: based solely on visual frames, and (2) visual-text: incorporating both visual frames and textual change caption. Further details of these strategies are presented in Appendix C.

**Sample.** Guided by the confidence score vector $\boldsymbol{w}$, we then sample a sparse subset of $k$ keyframes, $\mathcal{P}^s = \{I_1^s, I_2^s, \ldots, I_k^s\} \subset \mathcal{P}^{\text{FI}}$. This sampled set is prepended with $I_{\text{bef}}$ and appended with $I_{\text{aft}}$ to construct the procedure $\mathcal{P}$:

$$\mathcal{P} = \{I_{\text{bef}}, I_1^s, I_2^s, \ldots, I_k^s, I_{\text{aft}}\}. \tag{3}$$

This sequence serves as the input for the procedure modeling module, which learns to encode the dynamics of the change.

### 3.1.3 PROCEDURE MODELING MODULE

The core of our method is the procedure modeling module, designed to learn a rich, unified representation of the spatio-temporal dynamics within the procedure list $\mathcal{P}$. Inspired by Han et al. (2022), we employ a Transformer-based encoder as the backbone to model the change procedure, and utilize a pre-trained image tokenizer (Esser et al., 2021) to quantize $\mathcal{P}$ into discrete tokens, which serve as the targets for our masked multi-frame reconstruction objective. This encourages the model to infer missing spatio-temporal information, ensuring a deep understanding of the procedural dynamics.

**Input representation.** To prepare the encoder input, we first create a multi-modal sequence.

- **Visual stream:** Each frame in the sequence $\mathcal{P}$ (length $k+2$) is passed through a frozen CNN backbone (Esser et al., 2021) to extract a grid of $n_I$ patch-level visual features with $d$-dimensional vectors, yielding a sequential embeddings $\boldsymbol{e}^I \in \mathbb{R}^{(k+2)n_I \times d}$.
- **Textual stream:** Concurrently, the corresponding change caption $T$ is tokenized into $n_T$ tokens and embedded into a sequential embeddings $\boldsymbol{e}^T \in \mathbb{R}^{n_T \times d}$, where each token is embedded in a $d$-dimensional space.
- **Special tokens:** To structure this sequence for joint modeling, we prepend two learnable token embeddings: $\boldsymbol{e}^{\text{csy}} \in \mathbb{R}^d$ to $\boldsymbol{e}^I$ to capture frame consistency, and $\boldsymbol{e}^{\text{align}} \in \mathbb{R}^d$ to $\boldsymbol{e}^T$ to facilitate visual-textual alignment.

The concatenated input embeddings are $\{\boldsymbol{e}^{\text{align}}, \boldsymbol{e}^T, \boldsymbol{e}^{\text{csy}}, \boldsymbol{e}^I\}$.

**Multi-granularity masking.** To learn both coarse-grained semantics and fine-grained details, we introduce a multi-granularity masking strategy. This strategy is applied to the visual patch embeddings $\boldsymbol{e}^I$, while leaving the caption fully visible. This encourages the encoder to learn the underlying spatio-temporal dynamics by reconstructing masked regions under textual guidance. The strategy comprises four distinct masking schemes. The first operates at a coarse, frame-level granularity, while the remaining three focus on fine-grained, patch-level details:

- **Entire masking** masks the entire frame embeddings. It forces the encoder to reconstruct them using cross-modal context from the change caption.
- **Random patch masking** masks individual patches across the frames. This strategy encourages the encoder to learn distributed visual representations.
- **In-block masking** (Tan et al., 2021) masks a contiguous rectangular block of patches. This forces the encoder to learn the appearance and texture of local regions by "filling in" the masked area from its surrounding context.

- **Out-of-block masking** (Tong et al., 2022) masks all patches "outside" a specific block. This encourages the encoder to learn how to represent a region while understanding its relationship to the broader surrounding scene.

During each training step, every visual stream within the batch is independently masked using one of four randomly selected multi-granularity strategy. This selected strategy then generates a binary mask index set $\mathcal{M} \in \mathbb{R}^{(k+2)n_I}$, where a value of 1 indicates a patch to be masked. The chosen patch embeddings in $e^I$ are replaced with a learnable mask embedding $e_m \in \mathbb{R}^d$, creating the masked visual sequence $e_{\text{msk}}^I$. This sequence is then fed into the procedure encoder, which outputs a sequence of contextualized hidden states $H_{\text{msk}}$ for subsequent optimization:

$$H_{\text{msk}} = \{h^{\text{align}}, h^T, h^{\text{csy}}, h_{\text{msk}}^I\}. \tag{4}$$

Details on the formulation of these masking operations are provided in Appendix D.

### 3.1.4 OPTIMIZATION

The objective of procedure modeling, $\mathcal{L}_{\text{PRO}}$, comprises three components: (1) masked sequence modeling for reconstructing the masked regions in each frame, $\mathcal{L}_{\text{msm}}$; (2) cross-modal alignment between the visual frames and the textual change caption, $\mathcal{L}_{\text{align}}$; and (3) temporal consistency within the procedure sequence, $\mathcal{L}_{\text{csy}}$. The overall training objective is formulated as:

$$\mathcal{L}_{\text{PRO}} = \mathcal{L}_{\text{msm}} + \mathcal{L}_{\text{align}} + \mathcal{L}_{\text{csy}}. \tag{5}$$

**Masked sequence modeling.** We leverage a pre-trained image tokenizer (Esser et al., 2021) to tokenize each frame in procedure sequence $\mathcal{P}$ into $n_I$ discrete tokens. This tokenization process yields a corresponding discrete token sequence $z \in \mathbb{R}^{(k+2)n_I}$, serving as the ground truth. Given the modeled masked frame representations $h_{\text{msk}}^I$ from Eq. (4), we apply a linear projection layer as the masked sequence modeling head to map each position in the representation to a vocabulary-sized logits vector, yielding the predicted token sequence $y^{\text{msm}} \in \mathbb{R}^{(k+2)n_I}$. Given the masked frame embeddings $e_{\text{msk}}^I$ and the change caption $T$, the masked sequence modeling loss is defined as:

$$\mathcal{L}_{\text{msm}} = -\frac{1}{|\mathbf{I}_{\text{msk}}|} \sum_{i \in \mathbf{I}_{\text{msk}}} \log p(y_i^{\text{msm}} = z_i \mid e_{\text{msk}}^I, e^T), \tag{6}$$

where $\mathbf{I}_{\text{msk}} = \{i \mid \mathcal{M}_i = 1\}$ denotes the index set of positions masked in multi-granularity masking.

**Cross-modal alignment.** We incorporate an alignment loss to bridge the visual change procedure and its corresponding textual change caption. Using the special token representation $h^{\text{align}}$, which captures the relevance between visual and linguistic modalities as defined in Eq. (4), we optimize the encoder to effectively differentiate between aligned and non-aligned caption-procedure pairs:

$$\mathcal{L}_{\text{align}} = -\log p(1 \mid e_{\text{msk}}^I, e^T) - \log p(0 \mid e_{\text{msk}}^I, e^{\bar{T}}), \tag{7}$$

where $T$ is the change caption paired with frame embedding $e_{\text{msk}}^I$, and $\bar{T}$ is a negative sample not aligned with $e_{\text{msk}}^I$.

**Temporal consistency.** To mitigate the impact of temporal incoherence on the modeled procedure, we incorporate a consistency loss that encourages coherent representations across frames in the sequence. Using the special token representation $h^{\text{csy}}$ that captures frame consistency from Eq. (4), we optimize the encoder to differentiate between consistent and non-consistent frame sequences:

$$\mathcal{L}_{\text{csy}} = -\log p(1 \mid e_{\text{msk}}^I, e^T) - \log p(0 \mid e_{\text{msk}}^{\bar{I}}, e^T), \tag{8}$$

where $\bar{I}$ denotes the temporally warped version of $I$, intentionally disrupting the temporal consistency of the sequence. This warped version serves as a negative sample, encouraging the model to learn to distinguish between temporally coherent and incoherent sequences and generate more temporal coherent sequences. More details about constructing $\bar{I}$ are provided in Appendix E.

## 3.2 Implicit Procedure Captioning

The captioning stage employs an encoder-decoder architecture, with the encoder sharing weights with the procedure encoder from the prior stage. Directly leveraging the synthesized intermediate frames for captioning may introduce additional computational overhead as well as irrelevant noise. Therefore, we propose learnable procedure queries as dynamic "slots" inserted between start and end image features, which guide the encoder to implicitly infer change dynamics from the image pair. This design supports end-to-end training via captioning loss and yields encoder outputs that are both temporally coherent and task-relevant.

**Processing.** First, we process the image pair $(I_{\text{bef}}, I_{\text{aft}})$ with a CNN backbone to extract their respective visual patch features, $\boldsymbol{e}^{I_{\text{bef}}} \in \mathbb{R}^{n_I \times d}$ and $\boldsymbol{e}^{I_{\text{aft}}} \in \mathbb{R}^{n_I \times d}$. To bridge these two static representations, we introduce learnable procedure queries that replace the $k$ sampled intermediate frames within the previous stage. Since each frame is represented by $n_I$ patch features, we insert $k$ sets of queries, where each set contains $n_I$ learnable embeddings. This results in a total of $k \cdot n_I$ queries. Each of these queries is a learnable masked embedding ($\boldsymbol{e}_m$) used in the previous stage. The input sequence for the procedure encoder is constructed as follows:

$$\boldsymbol{e}^{\text{inp}} = \{\boldsymbol{e}^{I_{\text{bef}}}, \boldsymbol{e}_m, \cdots, \boldsymbol{e}_m, \boldsymbol{e}^{I_{\text{aft}}}\}. \tag{9}$$

The procedure encoder processes $\boldsymbol{e}^{\text{inp}}$ to produce representations that capture the underlining dynamic change procedure. Given these encoded representations, a Transformer-based textual decoder then learns to generate the change caption.

**Optimization.** The objective of captioning, $\mathcal{L}_{\text{CAP}}$, is an autoregressive language modeling loss that trains the entire model using ground-truth change captions as supervision:

$$\mathcal{L}_{\text{CAP}} = -\sum_i \log p(T_i \mid T_{<i}, \boldsymbol{e}^{\text{inp}}), \tag{10}$$

where $T_i$ denotes $i$-th word in the caption sequence.

## 3.3 ProCap Inference for Captioning

For inference on incoming image pair, the procedure encoder takes $\boldsymbol{e}^{\text{inp}}$ in Eq. (9) as the input. The output, a latent procedural representation, is then translated into the text caption by the textual decoder. Compared to other approaches, ProCap introduces only an additional $k \cdot n_I$ matrix for processing. As $n_I$ remains fixed throughout our experiments, the variation in computational overhead is primarily governed by $k$. With $k = 2$, this overhead is negligible, and we further analyze its effect in the following experiments.

## 4 Experiments

### 4.1 Datasets and Metrics

**Datasets.** We conduct experiments on three widely-used benchmark datasets: CLEVR-Change (Park et al., 2019), Spot-the-Diff (Jhamtani & Berg-Kirkpatrick, 2018), and Image-Editing-Request (Tan et al., 2019). These datasets cover a diverse range of change domains, from synthetic changes (CLEVR-Change) to subtle differences in natural scenes (Spot-the-Diff and Image-Editing-Request), allowing for a comprehensive evaluation of our model's capabilities. Additional details about dataset introduction are presented in Appendix G.

**Metrics.** To evaluate the quality of the generated captions, we report four standard metrics: BLEU-4 (B) (Papineni et al., 2002), METEOR (M) (Banerjee & Lavie, 2005), ROUGE-L (R) (Lin, 2004), and CIDEr (C) (Vedantam et al., 2015). All scores are obtained using the official Microsoft COCO evaluation toolkit (Chen et al., 2015). To evaluate the trade-off between captioning accuracy and our procedure modeling efficiency, we also measure inference efficiency in Tokens Per Second (TPS).

Table 1: Comparison with SOTA on CLEVR-Change, Spot-the-Diff and Image-Editing-Request.

| Methods | CLEVR-Change | | | | Spot-the-Diff | | | | Image-Editing-Request | | | |
|---|---|---|---|---|---|---|---|---|---|---|---|---|
| | B ↑ | M ↑ | R ↑ | C ↑ | B ↑ | M ↑ | R ↑ | C ↑ | B ↑ | M ↑ | R ↑ | C ↑ |
| *LLM-based Methods* | | | | | | | | | | | | |
| Qwen-VL (2023) | 48.9 | 36.0 | 71.2 | 119.8 | – | – | – | – | – | – | – | – |
| LLaVA-1.5 (2023) | 49.7 | 35.4 | 70.8 | 122.4 | – | – | – | – | – | – | – | – |
| VIXEN-C (2024) | – | – | – | – | – | – | – | – | 8.6 | 15.4 | 42.5 | 38.1 |
| FINER (2024) | **55.6** | **36.6** | **72.5** | **137.2** | **12.9** | **14.7** | **35.5** | **61.8** | 13.3 | 14.6 | 39.6 | 50.5 |
| LLaVA-1.5+RP (2025) | – | – | – | – | 9.7 | 13.0 | 30.8 | 43.2 | **16.2** | **19.5** | **46.7** | **60.9** |
| *Non-LLM-based Methods* | | | | | | | | | | | | |
| DUDA (2019) | 47.3 | 33.9 | – | 112.3 | 9.1 | 11.8 | 29.1 | 32.5 | 6.5 | 12.4 | 37.3 | 22.8 |
| DUDA+Aux (2021) | 51.2 | 37.7 | 70.5 | 115.4 | 8.1 | 12.4 | 31.3 | 38.1 | – | – | – | – |
| IFDC (2021) | 49.2 | 32.5 | 69.1 | 118.7 | 8.7 | 11.7 | 30.2 | 37.0 | – | – | – | – |
| NCT (2023b) | 55.1 | 40.2 | 73.8 | 124.1 | – | – | – | – | 8.1 | 15.0 | 38.8 | 34.2 |
| VARD-Trans (2023a) | 55.4 | 40.1 | 73.8 | 126.4 | – | – | – | – | 10.0 | 14.8 | 39.0 | 35.7 |
| SCORER+CBR (2023c) | 56.3 | 41.2 | 74.5 | 126.8 | 10.2 | 12.2 | – | 38.9 | 10.0 | 15.0 | 39.6 | 33.4 |
| MURAT+GCM (2024) | – | – | – | – | 10.2 | 13.1 | 33.1 | 39.4 | – | – | – | – |
| SMART (2024b) | 56.1 | 40.8 | 74.2 | 127.0 | – | 13.5 | 31.6 | 39.4 | 10.5 | 15.2 | 39.1 | 37.8 |
| DIRL+CCR (2024a) | – | – | – | – | 10.3 | 13.8 | 32.8 | 40.9 | 10.9 | 15.0 | 41.0 | 34.1 |
| RDD+ACR (2025) | 56.1 | 41.3 | 75.0 | 128.1 | 9.2 | 13.9 | 31.0 | **43.6** | – | – | – | – |
| MCT-CCDiff (2025) | **57.5** | 40.6 | **75.6** | 131.7 | 10.8 | 14.5 | 35.5 | 41.7 | 10.2 | 15.4 | 41.2 | 38.3 |
| **ProCap (Ours)** | 56.7 | **41.7** | 74.7 | **135.6** | 11.0 | 13.6 | 33.7 | 42.7 | 11.7 | 15.9 | 43.2 | 40.6 |

## 4.2 PERFORMANCE COMPARISON

### 4.2.1 BASELINES

We compare ProCap against a set of state-of-the-art methods, which are grouped into two categories: **1) Non-LLM-based methods** are the conventional paradigm, where a pre-trained CNN extracts visual features from input image pairs. These features are then fed into a Transformer-based encoder-decoder for captioning. We compare our method with: DUDA (Park et al., 2019), DUDA+Aux (Hosseinzadeh & Wang, 2021), IFDC (Huang et al., 2021), NCT (Tu et al., 2023b), VARD-Trans (Tu et al., 2023a), SCORER+CBR (Tu et al., 2023c), MURAT+GCM (Yue et al., 2024), SMART (Tu et al., 2024b), DIRL+CCR (Tu et al., 2024a), RDD+ACR (Li et al., 2025) and MCT-CCDiff (Hu et al., 2025). **2) LLM-based methods** leverage LLMs as powerful decoders, capitalizing on their vast knowledge and strong generative capabilities to improve caption quality. We compare our method with: Qwen-VL (Bai et al., 2023), LLaVA-1.5 (Liu et al., 2023), VIXEN-C (Black et al., 2024), FINER (Zhang et al., 2024) and LLaVA-1.5+RP (Jiao et al., 2025).

Our ProCap falls into the non-LLM-based category. While LLM-based methods benefit from rich prior knowledge, they typically entail heavy computation and large parameter sizes. In contrast, ProCap achieves strong performance with a lightweight, efficient architecture, avoiding reliance on large external language models.

### 4.2.2 RESULTS

We analyze ProCap's performance across three challenging scenarios, each testing a specific capability, in Table 1. Additional qualitative comparisons with state-of-the-art methods, extensive visualizations and case studies, are provided in Appendix M to illustrate ProCap effectiveness.

**Robustness to viewpoint changes.** First, we evaluate ProCap's robustness to viewpoint shifts on the CLEVR-Change dataset. As shown in Table 1, ProCap substantially outperforms all non-LLM methods on CIDEr and achieves competitive results on other metrics, indicating stronger semantic understanding. This improvement stems from our procedure modeling, which disentangles object transformations (the "what" of change) from camera movements (distractors) by analyzing the full transition path. Compared with LLM-based methods, ProCap surpasses Qwen-VL and LLaVA-1.5, and even outperforms FINER on most metrics, demonstrating strong reasoning capability without relying on large-scale decoders. A detailed comparison across different change categories is provided in Appendix I.

**Application to multiple changes in complex scenes.** Next, we evaluate ProCap on the Spot-the-Diff dataset, a more challenging real-world benchmark with cluttered scenes and multiple subtle changes. As shown in Table 1, ProCap achieves a competitive CIDEr score of 42.7. This demonstrates a key strength of our approach: by modeling change as a stepwise procedure, ProCap can "replay" the transformation process to disentangle concurrent changes and generate accurate captions. To better capture the rich dynamics in this setting, the frame interpolation module is pre-trained on a specialized video dataset (Oh et al., 2011) before ProCap's main training.

**Generalization to open-ended scenarios.** Finally, we assess ProCap's generalization abilities on the Image-Editing-Request dataset, which is characterized by its open-ended nature with largely unseen vocabulary. The results in Table 1 show that ProCap consistently outperforms all non-LLM baselines across all metrics. This suggests that by learning the "how" of a change (the procedure), our model develops a core understanding of the transformation itself, making it more resilient to variations in vocabulary and phrasing. While the LLM-based LLaVA-1.5+RP, with its vast knowledge base, still leads in overall accuracy, ProCap significantly narrows the performance gap. This demonstrates that procedural modeling is a powerful strategy for achieving robust generalization. It highlights a key distinction: whereas LLM-based methods obtain generalization by infusing external knowledge, ProCap's ability stems directly from its architectural innovation.

## 4.3 ABLATION STUDY

We study the impact of key components within the procedure modeling stage. Additional ablations on other components are detailed in Appendices K–L.

Table 2: Ablation study for explicit procedure modeling (EPM) and implicit procedure captioning (IPC) on CLEVR-Change dataset.

| EPM | IPC | $k$ | B ↑ | M ↑ | R ↑ | C ↑ |
|---|---|---|---|---|---|---|
| | | 0 | 47.2 | 35.8 | 68.6 | 108.4 |
| ✓ | | 0 | 52.6 | 38.0 | 70.1 | 112.7 |
| | ✓ | 1 | 47.3 | 36.3 | 68.8 | 106.2 |
| ✓ | ✓ | 1 | **56.5** | **41.9** | **75.5** | **128.5** |

Table 3: Effectiveness and performance comparison on CLEVR-Change dataset with varying procedure query set length $k$.

| Methods | $k$ | TPS ↑ | B ↑ | M ↑ | R ↑ | C ↑ |
|---|---|---|---|---|---|---|
| | 1 | 766.02 | 56.5 | 41.9 | **75.5** | 128.5 |
| **ProCap** | 2 | 699.04 | 56.7 | 41.7 | 74.7 | **135.6** |
| | 4 | 461.24 | **57.4** | **42.3** | **75.5** | 128.7 |
| | 7 | 270.55 | 56.8 | 41.8 | **75.5** | 130.5 |

**Impact of introducing explicit procedure modeling and implicit procedure captioning.** Table 2 analyzes the introduction of the explicit procedure modeling stage. We begin with a baseline encoder-decoder model trained on static image pairs from scratch. We then compare two enhancements to this baseline: (1) applying a pre-training stage (explicit procedure modeling), and (2) introducing a set of learnable procedure queries to enable implicit procedure captioning (implicit procedure captioning). Finally, we extend the model with both pre-training and learnable procedure queries. Compared to the baseline initialized randomly, applying the learnable queries directly (line 3) introduces random vectors of learnable queries, therefore lacking any temporal or procedural context. In this case, the model cannot effectively reason about the evolution from the "before" to the "after" image. Besides, applying explicit procedure modeling without the learnable queries (line 2) demonstrates that pre-training alone provides only limited gains, far smaller than the improvement observed when both pre-training and learnable queries are used together (line 4), with the CIDEr score significantly increasing to 128.5. This remarkable gain highlights our key insight: explicitly modeling the procedural dynamics of change is far more effective than simply comparing static image pairs. Notably, Table 14 in Appendix L further presents the advantages of implicit procedure captioning on reducing computational overhead and exhibiting greater robustness to visual noise, once the explicit modeling stage has provided the rich temporal understanding.

**Impact of procedure query set length $k$.** Table 3 shows the effect of varying the procedure query set length $k$ on both accuracy and computational efficiency (using one NVIDIA A40 GPU). Overall, efficiency decreases as the sequence length increases due to the heavier computational load. Considering the overall performance across the four evaluation metrics, C reaches its peak value of 135.6 at $k = 2$, while the other metrics exhibit a non-monotonic trend. Although the model achieves its best scores on B, M, and R at $k = 4$, the TPS drops substantially. Therefore, we select $k = 2$ as

it offers the optimal balance between capturing sufficient procedural detail for accuracy and maintaining computational efficiency. We further compare the performance and effectiveness of different procedure query set lengths with LLM-based methods in Appendix K.4.

Table 4: Ablation study for combinations of training objectives in explicit procedure modeling on CLEVR-Change and Spot-the-Diff.

| $\mathcal{L}_{\text{msm}}$ | $\mathcal{L}_{\text{align}}$ | $\mathcal{L}_{\text{csy}}$ | CLEVR-Change | | | | Spot-the-Diff | | | |
|---|---|---|---|---|---|---|---|---|---|---|
| | | | B ↑ | M ↑ | R ↑ | C ↑ | B ↑ | M ↑ | R ↑ | C ↑ |
| ✓ | | | 55.1 | 40.6 | 73.9 | 127.5 | 8.1 | 11.8 | 28.1 | 29.7 |
| ✓ | | ✓ | 55.5 | 40.6 | 73.8 | 127.1 | 7.9 | 11.7 | 28.0 | 28.9 |
| ✓ | ✓ | | 56.1 | 40.9 | 74.5 | 128.6 | 9.3 | 12.5 | 31.2 | 36.3 |
| ✓ | ✓ | ✓ | **56.7** | **41.7** | **74.7** | **135.6** | **11.0** | **13.6** | **33.7** | **42.7** |

**Integration of all the objectives.** Table 4 presents the contribution of each objective function from Eq. (5) within this stage. Building on the foundation of $\mathcal{L}_{\text{msm}}$, the full model—jointly optimized with all objectives—achieves peak performance, reaching a CIDEr score of 135.6 on the CLEVR-Change dataset and 42.7 on the Spot-the-Diff dataset. This corresponds to improvements of 8.5 on CLEVR-Change and 13.8 on Spot-the-Diff when removing $\mathcal{L}_{\text{align}}$, and gains of 7.0 and 6.4 on the two datasets when removing $\mathcal{L}_{\text{csy}}$. This further improvement highlights the integration of the other two losses, each targeting a specific aspect of the procedure representation: The alignment loss ($\mathcal{L}_{\text{align}}$) acts as a crucial bridge, grounding the visual procedure representation in the linguistic domain. It explicitly enforces that the learned procedure is not just visually coherent, but also semantically aligned with its corresponding textual description. Meanwhile, the consistency loss ($\mathcal{L}_{\text{csy}}$) ensures the temporal order of the procedure, specifically penalizing temporally incoherent (e.g., shuffled) sequences. This forces the model to be sensitive to the correct order of events within the change.

## 5 CONCLUSION

In this paper, we introduce ProCap, a novel two-stage paradigm that shifts change captioning from modeling static image comparison to the dynamic change procedure. The first stage learns a procedure encoder that models change dynamics by performing caption-conditioned masked reconstruction on a sparse set of intermediate frames, distilled from the synthesized explicit procedure. The second stage, captioning, introduces efficient and learnable procedure queries to represent the implicit process within the image pair. This design enables end-to-end training without costly intermediate frame synthesis during inference. Experiments across diverse datasets demonstrate ProCap effectiveness.

## ACKNOWLEDGEMENTS

This work is supported by the National Natural Science Foundation of China under Grants 62476188, the National Key R&D Program of China (No. 2022ZD0160601), and the Key Laboratory of Computing Power Network and Information Security, Ministry of Education under Grant No.2024PY024.

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

## A  APPENDIX OVERVIEW

The appendix provides the following details:

## B  RELATED WORK

### B.1  FRAME INTERPOLATION

Frame Interpolation (FI) aims to synthesize a dynamic visual transition between a given start and end frame. Existing approaches have achieved remarkable progress with powerful generative models, including denoising diffusion models that generate intermediate frames from noise (Voleti et al., 2022; Höppe et al., 2022; Pallotta et al., 2025; Zhang et al., 2025b; Hur et al., 2025) and Transformer-based architectures that predict missing content autoregressively (Yan et al., 2021; Ge et al., 2022). A notable solution is text-conditioned interpolation (Han et al., 2022; Fu et al., 2023), which uses textual descriptions to guide the synthesis in a controllable manner. However, existing FI research primarily focuses on generating visually realistic videos, rather than supporting reasoning for downstream tasks such as captioning. To enhance change captioning, we draw inspiration from FI techniques to explicitly synthesize a procedural sequence and model the underlying change dynamics, thus providing a richer foundation for downstream reasoning.

## C  SEMANTIC SIMILARITY FUNCTION

To quantify the informativeness of the intermediate frame, we investigate two strategies for computing the similarity metric, $s(\cdot, \cdot)$ in Eq. (2) in the main paper. These strategies are defined by the modalities they incorporate: (1) visual-only, which relies solely on visual frame information, and (2) visual-text, which integrates both visual frames and the corresponding textual change caption. The effectiveness of these strategies is experimentally presented in Appendix K.2.

## C.1 VISUAL-ONLY

Modeling fine-grained semantic similarity in images—a task that requires detailed comparison of object attributes and context—poses a challenge for conventional feature extractors. Extractors like ResNet (He et al., 2016), which are pre-trained on classification tasks, tend to produce coarse, global feature representations that overlook subtle semantic distinctions. To capture them, we employ DINOv2 (Oquab et al., 2023), a powerful Vision Transformer (ViT) (Dosovitskiy et al., 2021) pre-trained through self-supervision. Its attention-based architecture and training objective encourage the extraction of features that are highly sensitive to local details and object-level semantics. Consequently, we employ DINOv2 to extract features from each image and compute their cosine similarity, providing a robust measure of their semantic alignment.

We formalize visual similarity using features from a pretrained DINOv2 model with a ViT-L/14 backbone, denoted as the encoder $\mathcal{E}^{\text{DINO}}(\cdot)$. Given a target image $I_t$ (where $t \in \{\text{bef}, \text{aft}\}$) and the generated frame set $\mathcal{P}^{\text{FI}}$, we define the visual similarity score set $s_{\text{vis}}(I_t, \mathcal{P}^{\text{FI}})$ as:

$$s_{\text{vis}}(I_t, \mathcal{P}^{\text{FI}}) = \{s(I_t, I_i) \mid I_i \in \mathcal{P}^{\text{FI}}\}, \tag{11}$$
$$s(I_t, I_i) = \text{sim}[\mathcal{E}^{\text{DINO}}(I_t), \mathcal{E}^{\text{DINO}}(I_i)],$$

where $\text{sim}[\cdot, \cdot]$ represents the cosine similarity between the extracted features.

## C.2 VISUAL-TEXT

While visual similarity with $\mathcal{P}^{\text{FI}}$ serves to measure information redundancy, it is insufficient for verifying the semantic correctness of the change transformation. A purely visual metric is text-agnostic; thus, a pseudo-frame can be a visually plausible interpolation yet fail to represent the specific change conveyed by the ground-truth caption. To resolve this issue and enforce semantic validity, we incorporate the ground-truth change caption to explicitly model the informativeness of each pseudo-frame.

To this end, we employ the pretrained CLIP-based model from Guo et al. (2022), which is specifically designed to measure semantic alignment between an image-pair transformation and a textual description. The model provides a dedicated image-pair encoder $\mathcal{E}_I^{\text{CLIP}}(\cdot, \cdot)$, and a text encoder $\mathcal{E}_T^{\text{CLIP}}(\cdot)$. The similarity function $s_{\text{vis-text}}(\cdot, \cdot)$ between a target image $I_t$ and pseudo-frame candidates $\mathcal{P}^{\text{FI}}$ under caption $T$ is defined as:

$$s_{\text{vis-text}}(I_t, \mathcal{P}^{\text{FI}} \mid T) = \{s(I_t, I_i, T) \mid I_i \in \mathcal{P}^{\text{FI}}\}, \tag{12}$$
$$s(I_t, I_i, T) = \text{sim}[\mathcal{E}_I^{\text{CLIP}}(I_t, I_i), \mathcal{E}_T^{\text{CLIP}}(T)],$$

where $T$ is the change caption corresponding to the image pair $(I_{\text{bef}}, I_{\text{aft}})$. If a pseudo-frame $I_i$ is semantically misaligned with the caption $T$, it will receive a lower similarity score, indicating that it contains incorrect or irrelevant information about the change transformation.

## D MULTI-GRANULARITY MASKING SCHEMES

We adopt four masking strategies as illustrated in Figure 3 during the training of Explicit Procedure Modeling: (1) entire masking, (2) random patch masking, (3) in-block masking (Tan et al., 2021) and (4) out-of-block masking (Tong et al., 2022). During training, one masking strategy is randomly selected with a probability of 0.1, 0.7, 0.1, 0.1, respectively, and applied to each sample in a batch. Given an input image embedding $e^I \in \mathbb{R}^{(k+2)n_I \times d}$, the binary mask index set is denoted as $\mathcal{M} \in \mathbb{R}^{(k+2)n_I}$, where a value of 1 on index $i$ indicates the $i$-th patch to be masked.

**Entire Masking.** This strategy masks all embeddings in the process sequence, forcing the model to reconstruct the entire process solely based on the accompanying text sequences in the alignment setting. Formally, the masking probability is defined as:

$$p(e_i^I = e_{\text{msk}}^I \mid e_i^I \in e^I) = 1. \tag{13}$$

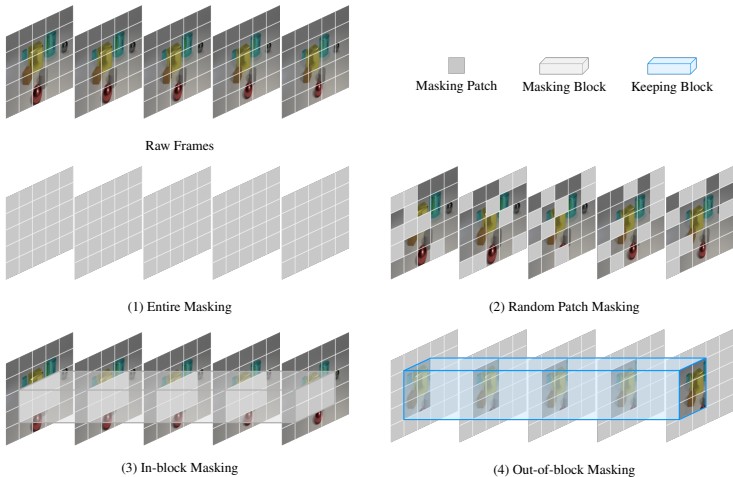

Figure 3: Four masking schemes in the proposed multi-granularity strategy. We mask visual patch embeddings for reconstruction during training; the masks are visualized at the patch level for clarity.

**Random Patch Masking.** Given an interval $(a, b)$, the masking probability for index $i$ is sampled from a uniform distribution over this interval, denoted as $\mathcal{U}(a, b)$, where $a$ and $b$ are set to 0.2 and 0.5 respectively in experiments. Specifically, for $e_i^I \in e^I$, the probability of replacing $e_i^I$ with a mask token $e_{\text{msk}}^I$ is given by:

$$p(e_i^I = e_{\text{msk}}^I \mid e_i^I \in e^I) = p_i \quad \text{where } p_i \sim \mathcal{U}(a, b). \tag{14}$$

**In-block and Out-of-block Masking.** Given an unflattened image embedding $e_k^I \in \mathbb{R}^{h \times w \times d}$, a rectangular region, whose area ratio to the whole image is randomly sampled within $[0.2, 0.8]$ (with an expected value of approximately 0.5), is randomly selected with bottom-left corner at $(x_1, y_1)$ and top-right corner at $(x_2, y_2)$:

$$\mathcal{R} = \{(i, j) \mid x_1 \leq i \leq x_2, y_1 \leq j \leq y_2\}, \tag{15}$$

where $0 < x_1 < x_2 < w$ and $0 < y_1 < y_2 < h$. In In-block masking, all embeddings within this region are masked:

$$p\left(e_k^{m,n} = e_{\text{msk}} \mid e_k^{m,n} \in e_k^I, \ k \in \{1, ..., k+2\}, \ (m, n) \in \mathcal{R}\right) = 1. \tag{16}$$

Conversely, in Out-of-block masking, all embeddings outside the selected region are masked:

$$p\left(e_k^{m,n} = e_{\text{msk}} \mid e_k^{m,n} \in e_k^I, \ k \in \{1, ..., k+2\}, \ (m, n) \notin \mathcal{R}\right) = 1. \tag{17}$$

## E  WARPING STRATEGIES

We apply four widely used warping strategies for disrupting the temporal consistency of the frame sequence for training in Explicit Procedure Modeling stage: (1) batch procedure frame shuffle, (2) frame shuffle, (3) color shifting, and (4) affine transformation.

**Batch Procedure Frame Shuffle.** Given a batch of procedures $\{\mathcal{P}_1, \mathcal{P}_2, ..., \mathcal{P}_B\}$, the sequence frame shuffle strategy randomly selects two positions $i$ and $j$ from two different procedure $\mathcal{P}_{b_1}$ and $\mathcal{P}_{b_2}$, respectively. It then replaces the frame $I_i \in \mathcal{P}_{b_1}$ with $I_j \in \mathcal{P}_{b_2}$.

**Frame Shuffle.** Given a procedure $\mathcal{P}$, a random permutation is applied to its frames to produce a shuffled sequence $\mathcal{P}'$, which serves as the augmented data.

**Color Shifting.**   Given a procedure $\mathcal{P} \in \mathbb{R}^{T \times H \times W \times 3}$, we randomly select a single RGB channel and add a random scalar value $a$ to all the pixels in that channel across the entire sequence. This results in a color shifting augmentation:

$$I_{\text{shift}}^c = I^c + a, \tag{18}$$

where $I^c \in \mathbb{R}^{T \times H \times W}$ represents the selected RGB channel of all images in $\mathcal{P}$.

**Affine Transformation.**   We apply a random affine transformation to the input image $I_i \in \mathcal{P}$. Specifically, we sample:

- a rotation angle $\theta \sim \mathcal{U}(-\alpha, \alpha)$,
- horizontal and vertical transitions $t_x, t_y \sim \mathcal{U}(-\tau, \tau)$,
- and a scaling factor $s \sim \mathcal{U}(1 - \mu, 1 + \mu)$.

where $\alpha$, $\tau$ and $\mu$ are user-defined hyperparameters, which is set to 30, 0.1 and 0.1 in out experiments respectively. An affine transformation matrix is defined as:

$$\boldsymbol{A} = \begin{bmatrix} s \cdot \cos\theta & -s \cdot \sin\theta & t_x \\ s \cdot \sin\theta & s \cdot \cos\theta & t_y \end{bmatrix}. \tag{19}$$

For each position of the input image $[x \ y]$, the augmented output can be denoted as:

$$\begin{bmatrix} x' \\ y' \end{bmatrix} = \begin{bmatrix} s \cdot \cos\theta & -s \cdot \sin\theta \\ s \cdot \sin\theta & s \cdot \cos\theta \end{bmatrix} \begin{bmatrix} x \\ y \end{bmatrix} + \begin{bmatrix} t_x \\ t_y \end{bmatrix}. \tag{20}$$

# F   ASYMPTOTIC UPPER BOUND

In this section, we will discuss the asymptotic upper bound in inference. Let $n_I$ denote the length of image embeddings and $n_T$ the length of text embeddings. For simplicity, we assume all embeddings have a uniform dimensionality $d$. The asymptotic upper bound of the entire model in inference can be divided into two compnents: one corresponding to procedure encoder, and the other to the text decoder.

**Analysis for Procedure Encoder.**   Before sending to the Transformer-based procedure encoder, image pairs are first encoded into embeddings via a CNN. These embeddings are then concatenated with masked embeddings to form a sequence of shape $(k + 2)n_I \times d$. For clarity in complexity analysis, we let $K = k + 2$ and denote $K \cdot n_I$ as $n_{\mathcal{P}}$. The time complexity of CNN can be denoted as $O(n_I \times \text{channels}^2 \times \text{kernels})$. Assuming a constant kernel size and fixed number of channels, the time complexity of a convolutional layer scales linearly with the number of output pixels, i.e., $O(n_I)$. For each layer in the Transformer architecture, the input embeddings are linearly projected to obtain queries, keys, and values, incurring a time complexity of $O(n_{\mathcal{P}} \times d^2)$. The self-attention mechanism, as introduced by Vaswani et al. (2017), computes attention as follows:

$$\text{Attention}(Q, K, V) = \text{softmax}\left(\frac{QK^\top}{\sqrt{d}}\right)V. \tag{21}$$

This step dominates the computational cost of the attention mechanism, with a time complexity of $O(n_{\mathcal{P}}^2 \times d)$.

As a result, the final asymptotic upper bound of procedure encoder of $l_e$ layers can be denoted as:

$$O(n_I + l_e \times (n_{\mathcal{P}} \times d^2 + n_{\mathcal{P}}^2 \times d)). \tag{22}$$

Given that $d \gg 1$, the lower-order term $O(n_I)$ becomes negligible, and the complexity can be approximated by:

$$O(l_e \times (n_{\mathcal{P}} \times d^2 + n_{\mathcal{P}}^2 \times d)). \tag{23}$$

**Analysis for Text Decoder.** The text decoder for captioning is a $l_d$-layer Transformer decoder, which includes both self-attention and cross-attention mechanisms. For the self-attention mechanism, the time complexity per layer is given by $O(n_T \times d^2 + n_T^2 \times d)$. For the cross-attention mechanism, where attention is computed between the change procedure sequence and the text sequence, the time complexity can be expressed as:

$$O(n_\mathcal{P} \times n_T \times d + n_\mathcal{P} \times d^2 + n_T \times d^2), \tag{24}$$

accounting for the projections of both input sequences and the attention computation. As $n_\mathcal{P} \gg n_T$ and $d \gg N_c$ in our experiments, Eq. (24) can be approximated by:

$$O(n_\mathcal{P} \times d + n_\mathcal{P} \times d^2 + d^2) \approx O(n_\mathcal{P} \times d^2). \tag{25}$$

Therefore, the final asymptotic upper bound of a $l_d$-layer text decoder can be denoted as:

$$O(l_d \times (n_T \times d^2 + n_T^2 \times d + n_\mathcal{P} \times d^2)), \tag{26}$$

which can be approximated by:

$$O(l_d \times n_\mathcal{P} \times d^2). \tag{27}$$

**Asymptotic Upper Bound in Inference.** Comprising Eq. (23) and Eq. (27), the final asymptotic upper bound of the entire model in inference can be denoted as:

$$O(l_e \times (n_\mathcal{P} \times d^2 + n_\mathcal{P}^2 \times d) + l_d \times n_\mathcal{P} \times d^2). \tag{28}$$

Since $l_d$ is a small constant, it can be omitted from the asymptotic expression. Therefore, the asymptotic upper bound can be denoted as:

$$O(l_e \times (n_\mathcal{P} \times d^2 + n_\mathcal{P}^2 \times d)). \tag{29}$$

Substituting $n_\mathcal{P} = K \times n_I$ into the above expression yields:

$$O(l_e \times (K \times n_I \times d^2 + K^2 \times n_I^2 \times d)). \tag{30}$$

It can be noted that the inference computation scales quadratically with respect to procedure length $K$. Therefore, it is necessary to reach a balance between performance and inference computation cost.

## G  INTRODUCTION OF DATASETS

We conduct experiments on three widely used benchmark datasets: Spot-the-Diff (Jhamtani & Berg-Kirkpatrick, 2018), CLEVR-Change (Park et al., 2019), and Image-Editing-Request (Tan et al., 2019). In this section, we provide a detailed overview of each dataset.

**Spot-the-Diff** is the first dataset specifically designed for change captioning. It is constructed by sampling from VIRAT (Oh et al., 2011), a realistic video surveillance dataset. The dataset comprises 13,192 pairs of similar images, each paired with a human-annotated change caption. Since the image pairs are derived from surveillance videos, they are well-aligned, and each pair contains at least one semantic change. The dataset is split to training, validation and testing sets with an 8:1:1 distribution.

**CLEVR-Change** is a synthetic dataset generated using CLEVR (Johnson et al., 2017), a rendering engine capable of producing images of objects with complex relationships. It consists of 79,606 pairs of similar images with 493,735 change caption annotations, which is split into 67,660, 3,976, and 7,970 training/validation/test image pairs, respectively. Unlike Spot-the-Diff, CLEVR-Change introduces distractors alongside semantic changes—for example, variations in viewpoint that do not alter object positions. This design poses greater challenges for change captioning, requiring models to distinguish genuine semantic changes from irrelevant visual differences and to be more robust in reasoning about visual transformations.

**Image-Editing-Request** provides similar image pairs with image editing approaches guided by instructions. It comprises 3,939 similar image pairs with 5,695 human-annotated instructions as change captions. The dataset is segmented into 3,061 training pairs, 383 validation pairs, and 495 testing pairs.

Table 5: Evaluation on CLEVR-Change with varied change categories by METEOR.

| Method | Color | Texture | Add | Drop | Move |
|---|---|---|---|---|---|
| DUDA (2019) | 32.8 | 27.3 | 33.4 | 31.4 | 23.5 |
| DUDA+Aux (2021) | 36.1 | 30.4 | 37.8 | 36.7 | 27.0 |
| IFDC (2021) | 33.1 | 27.9 | 36.2 | 31.4 | 31.2 |
| NCT (2023b) | 39.1 | 36.3 | 39.0 | 37.2 | 30.5 |
| SMART (2024b) | 40.2 | 37.8 | 39.3 | 38.1 | 31.5 |
| DIRL+CCR (2024a) | **40.7** | **38.2** | 40.0 | 37.9 | 33.5 |
| **ProCap (Ours)** | 39.7 | 37.6 | **41.0** | **39.0** | **38.1** |

Table 6: Extended comparison with MCT-CCDiff on the Spot-the-Diff dataset, where $^\dagger$ denotes model training with LLM-augmented captions.

| Method | Speed (s/caption) ↓ | B ↑ | M ↑ | R ↑ | C ↑ |
|---|---|---|---|---|---|
| MCT-CCDiff (2025) | 0.91 | 10.8 | **14.5** | **35.5** | 41.7 |
| **ProCap (Ours)** | 0.04 | 11.0 | 13.6 | 33.7 | 42.7 |
| **ProCap$^\dagger$ (Ours)** | 0.04 | **11.7** | 14.2 | 34.6 | **44.6** |

## H  IMPLEMENTATION DETAILS

We employ a pre-trained frame interpolation model, VFIformer (Lu et al., 2022), to synthesize pseudo change procedures, with the process length set as $l = 7$. To balance captioning quality and inference efficiency, we sample $k = 2$ intermediate frames. For image representation, we fine-tune a pre-trained VQGAN on the change captioning datasets via an image reconstruction task. The VQGAN is configured with a codebook size of $K = 1024$ and a latent dimension $d_z = 256$. Input images are resized to $224 \times 224$ and encoded into a latent resolution of $14 \times 14$. The procedure encoder is configured with $l_e = 12$ layers on CLEVR-Change and Image-Editing-Request, and $l_e = 4$ layers on Spot-the-Diff. The hidden size is fixed at 768. The caption decoder consists of $l_d = 2$ layers on CLEVR-Change and Image-Editing-Request datasets, and consists of $l_d = 3$ layers on Spot-the-Diff dataset, with a common hidden size of 512.

In the Explicit Procedure Modeling stage, we train our model for 200,000 steps on 2 NVIDIA A40 GPUs using a warm-up strategy that linearly increases the learning rate from $1 \times 10^{-6}$ to $1 \times 10^{-4}$ over the first 5,000 steps. The total batch size is set to 8. In the Implicit Procedure Captioning stage, we train our model for 40 epochs with the total batch size of 16 on 1 NVIDIA A40 GPU. The procedure encoder is optimized with a fixed learning rate of $5 \times 10^{-5}$ on the CLEVR-Change and Image-Editing-Request datasets, and $2 \times 10^{-5}$ on the Spot-the-Diff dataset. Meanwhile, the caption decoder adopts a warm-up schedule that linearly increases the learning rate from 0 to $5 \times 10^{-5}$ during the first 10% of total training steps for all datasets.

Code and data for our experiments will be made publicly available.

## I  COMPARISON ON VARIED CHANGE CATEGORIES

In this section, we present a detailed comparison of performance across different change categories on CLEVR-Change, evaluated with METEOR against SOTA methods. Table 5 shows that our approach achieves competitive results on color and texture changes, and attains the best performance on addition, removal, and movement changes. In particular, it significantly outperforms the current SOTA method on movement changes, indicating a superior ability to distinguish action-related changes in the presence of environmental distractors.

## J  EXTENDED COMPARISON WITH MCT-CCDIFF

To better understand the performance characteristics of ProCap on the Spot-the-Diff dataset, we conducted an extended analysis comparing our method with the current SOTA approach, MCT-

CCDiff (Hu et al., 2025). We observed that MCT-CCDiff reports notably higher METEOR and ROUGE scores on this dataset, while ProCap achieves superior CIDEr performance. Upon examination, we found that this discrepancy is primarily attributable to differences in the richness of the training captions rather than limitations of the model architecture itself.

As documented in MCT-CCDiff, their training pipeline expands the original Spot-the-Diff training set with GPT-generated captions, substantially enriching the linguistic diversity of the supervision. In contrast, our primary experiments strictly follow the original, unaugmented annotations. Since METEOR and ROUGE are highly sensitive to caption diversity and surface-level phrasing, this difference in training data preparation naturally affects these metrics.

To isolate the effect of caption richness, we conducted an additional experiment in which we augmented the Spot-the-Diff training captions using Qwen3 (Yang et al., 2025), following the strategy introduced in MCT-CCDiff. As shown in Table 6, under this matched setting, ProCap achieves comparable METEOR and ROUGE scores and surpasses MCT-CCDiff on the more semantically aligned measures, including CIDEr and BLEU-4 (with improvements of +7% and +8%, respectively). These results indicate that the gap previously observed on METEOR and ROUGE stems largely from the linguistic properties of the training set rather than from the robustness of the model.

In addition to accuracy, we also compare inference efficiency. Under identical conditions, ProCap is 22× faster than MCT-CCDiff while maintaining superior CIDEr performance. This demonstrates that ProCap offers not only competitive captioning quality but also a significantly better efficiency–effectiveness trade-off compared to existing non-LLM SOTA approaches.

## K    ABLATION ON EXPLICIT PROCEDURE MODELING

This section extends the ablation study from Sec. 4.3 of the main paper with a detailed component analysis on the three commonly used datasets. We specifically evaluate the contributions of individual components within the Procedure Generation, Confidence-based Frame Sampling, and Procedure Modeling Modules.

### K.1    MORE ABLATION ON SPOT-THE-DIFF DATASET

Tables 7 and 8 present additional ablation studies on the Spot-the-Diff dataset, which contains more realistic scenarios compared with CLEVR-Change. Consistent patterns emerge across these experiments, further demonstrating the effectiveness of our method and its strong generalization ability in real-world settings.

Table 7: Ablation study for explicit procedure modeling (EPM) and implicit procedure captioning (IPC) on Spot-the-Diff dataset.

| EPM | IPC | $k$ | B ↑ | M ↑ | R ↑ | C ↑ |
|---|---|---|---|---|---|---|
| | | 0 | 7.9 | 11.7 | 28.0 | 28.9 |
| ✓ | | 0 | 8.5 | 12.1 | 27.8 | 30.6 |
| | ✓ | 1 | 8.3 | 12.1 | 27.5 | 29.8 |
| ✓ | ✓ | 1 | **8.6** | **12.5** | **32.2** | **36.0** |

Table 8: Effectiveness and performance comparison on Spot-the-Diff dataset with varying procedure query set length $k$.

| Methods | $k$ | B ↑ | M ↑ | R ↑ | C ↑ |
|---|---|---|---|---|---|
| | 1 | 8.6 | 12.5 | 32.2 | 36.0 |
| ProCap | 2 | **11.0** | **13.6** | **33.7** | **42.7** |
| | 4 | 8.5 | 12.4 | 27.7 | 31.3 |
| | 7 | 7.5 | 11.8 | 25.7 | 29.2 |

### K.2    PROCEDURE GENERATION MODULE

We investigate the interaction between the number of generated pseudo-frames, $l$, and the choice of semantic similarity function for keyframe sampling. To this end, we evaluate the two functions (see Appendix C) within our Confidence-based Frame Sampling Module, benchmarking them against a random sampling baseline that selects frames uniformly.

**Varying number of generated pseudo-frames $l$.** Figure 4 examines how varying the number of generated frames $l$ affects captioning performance, while keeping the number of sampled keyframes in the Procedure Modeling Module fixed at $k = 2$. The results highlight a clear trade-off: increasing $l$ enriches spatio-temporal cues but simultaneously introduces substantial redundancy and noise.

This trade-off is most pronounced in the random sampling strategy on the CLEVER-Change dataset and in the visual-only sampling strategy on the Spot-the-Diff dataset. In both cases, performance improves as $l$ increases from 3 to 7, but then noticeably degrades when $l$ rises to 15. Although our proposed sampling strategy also experiences a slight decline on the Spot-the-Diff dataset as $l$ continues to grow, it consistently outperforms the other two strategies. This suggests that, without semantic guidance, redundant and irrelevant frames can easily overwhelm the model, reinforcing the need for more robust sampling mechanisms capable of isolating truly informative temporal cues while filtering out misleading ones. Based on these observations, we set $l = 7$ as the default configuration in our experiments.

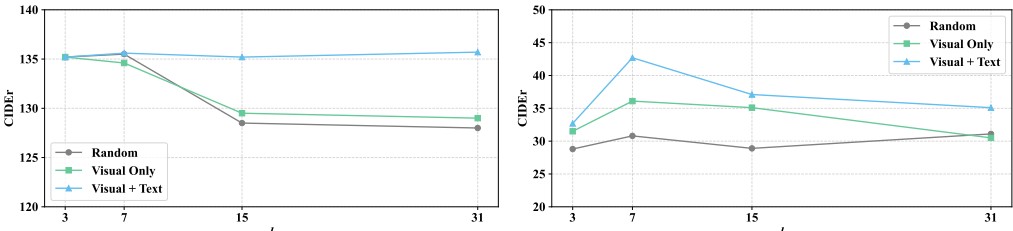

Figure 4: Comparison of CIDEr scores across four sampling strategies with respect to the number of pseudo-frames $l$ on CLEVR-Change dataset (left) and Spot-the-Diff dataset (right). Each strategy is set to sample two key frames from the pseudo-frames.

**Measure of constraint in FI model.** As defined in Sec. 3, we formalized the change procedure as a mapping $\gamma_T : [0, 1] \rightarrow \mathcal{I}$, where $T$ is a referred change caption and $\mathcal{I}$ denotes the space of all possible images. As the mapping is non-bijective, without additional constraints, there exist infinite procedures for the same image pair. In our experiment, to restrict the solution space, we adopt an off-the-shelf optical-flow-based frame interpolation method to synthesize change procedures, where the optical flow serves as a strong constraint: the intermediate frame is obtained by warping the before and after images according to the linearly interpolated optical flow, rather than being generated from scratch. To empirically demonstrate the necessity of these constraints, we compared our approach on the Image-Editing-Request dataset with one diffusion-based frame interpolation, which operates within a significantly less constrained solution space. As shown in Table 9, relaxing the constraints leads to noticeable performance degradation compared to the optical-flow-based approach. We attribute this drop to the stochastic nature of diffusion models. Unlike optical-flow-based methods that enforce strict pixel-wise correspondence, diffusion models inherently introduce unpredictable and uncontrollable visual variations in the intermediate frames (as shown in Figure 5). These unintended variations make procedure modeling more difficult, hindering effective model training.

Table 9: Performance comparison with different constraints of the FI model on the Image-Editing-Request dataset.

| FI Models | B ↑ | M ↑ | R ↑ | C ↑ |
|---|---|---|---|---|
| Ours (diffusion-based (2025a)) | 9.9 | 15.3 | 41.3 | 37.8 |
| Ours (optical-flow-based (2022)) | 11.7 | 15.9 | 43.2 | 40.6 |

### K.3 CONFIDENCE-BASED FRAME SAMPLING MODULE

**Impact of semantic similarity functions.** Figure 4 illustrates the comparative performance of three distinct semantic similarity functions for keyframe selection. Our analysis yields the following observations. **(1) Random Sampling vs Visual Only Strategies:** Compared with random sampling, *Visual Only* demonstrates benefits, particularly when sampling a larger number of pseudo-frames, such as $l = 15$. This highlights the effectiveness of filtering out redundant frames in long frame sequences. However, *Visual Only* strategy still exhibits a clear performance decline as $l$ increases, indicating its sensitivity to irrelevant visual content when textual grounding is absent. **(2) Visual+Text Strategy:** In contrast, *Visual+Text* strategy consistently outperforms other strategies

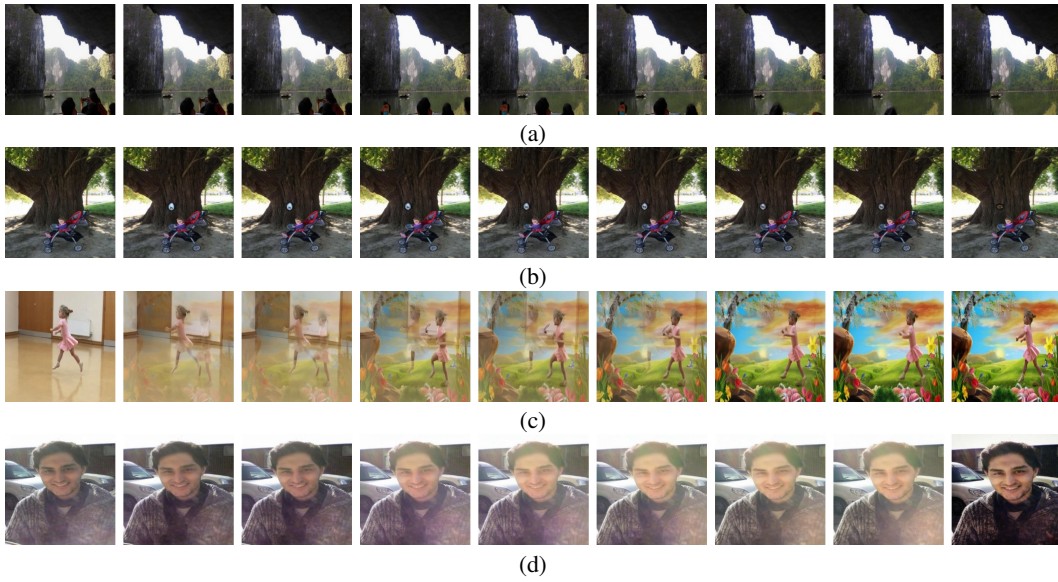

Figure 5: Uncontrollable predicted intermediate frames examples of diffusion-based FI models. Samples (a) and (b) show an unexpected object prediction, while samples (c) and (d) show an unexpected motion generation.

across most evaluated values of $l$. Its performance remains robust even as $l$ increases, suggesting that the integration of textual cues provides a strong guiding signal for identifying informative and relevant frames. This makes *Visual+Text* strategy resilient to noisy or redundant frames within the temporal sequence. **(3) Overall:** These results collectively highlight the effectiveness of leveraging multimodal signals—particularly textual grounding—for key frame selection under varying temporal resolutions. As a result, we select *Visual+Text* strategy for our model.

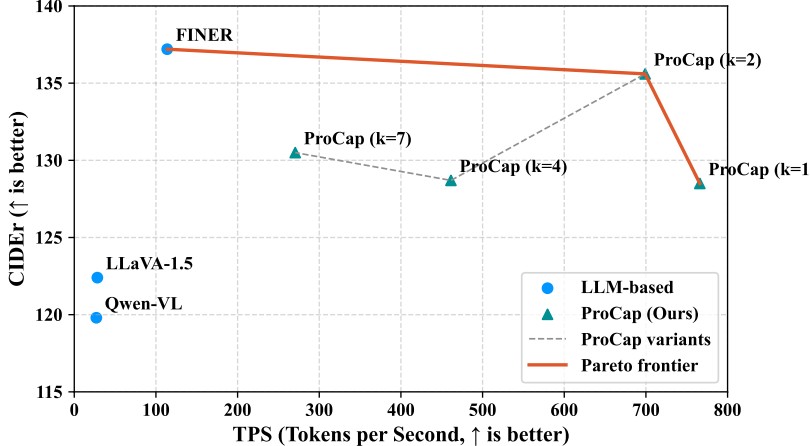

Figure 6: Effectiveness and performance comparison with LLM-based methods on CLEVR-Change dataset.

### K.4 PROCEDURE MODELING MODULE

**Comparison with LLM-based methods on different query set lengths $k$.** Figure 6 presents the performance comparison with LLM-based approaches on the CLEVR-Change dataset, using the same query set configurations as in Sec. 4.3. Our method achieves clear improvements over general multi-modal large language models Qwen-VL and LLaVA-1.5, demonstrating its strong capabil-

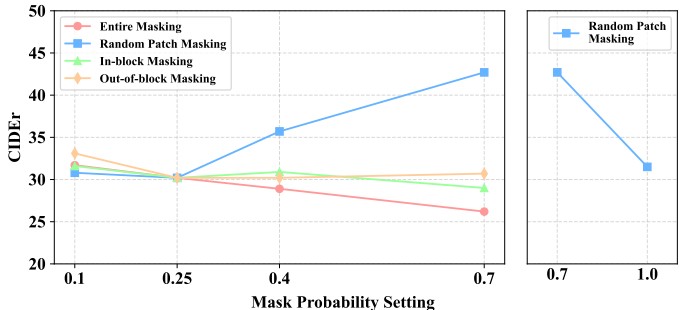

Figure 7: Comparison of CIDEr scores on the Spot-the-Diff dataset under different masking strategies across varying probability settings. When the probability of one strategy is set to $p$, the probabilities of the remaining three strategies are each set to $(1-p)/3$.

ity in change captioning. Although the specifically trained LLM-based method FINER performs well on CLEVR-Change, it suffers from substantial computational cost due to its large number of parameters. In contrast, our approach attains competitive overall performance while maintaining remarkable effectiveness at $k = 2$.

**Impact of caption-conditioning.**    Table 10 presents the benefit of incorporating ground-truth captions as a condition during procedure modeling. A significant performance boost is observed when the model is conditioned on the text, compared to using visual inputs alone. This shows the power of cross-modal learning in our procedure modeling. The caption acts as a powerful semantic prior, achieving two key objectives: (1) helping understand the nature of visual changes, and (2) achieving an early alignment between visual dynamics and linguistic contents. By learning to generate a procedure that is consistent with the target description, the model produces a representation that is not only visually coherent but also semantically aligned with the captioning stage. Therefore, we utilize ground-truth captions as conditional guidance for training the procedure modeling module.

Table 10: Ablation study for caption-conditioning in explicit procedure modeling on CLEVR-Change and Spot-the-Diff.

| Settings | CLEVR-Change | | | | Spot-the-Diff | | | |
|---|---|---|---|---|---|---|---|---|
| | B ↑ | M ↑ | R ↑ | C ↑ | B ↑ | M ↑ | R ↑ | C ↑ |
| w/o caption | 57.0 | 40.9 | 74.7 | 128.8 | 8.0 | 11.6 | 28.1 | 28.9 |
| w/ caption | 56.7 | 41.7 | 74.7 | 135.6 | 11.0 | 13.6 | 33.7 | 42.7 |

Table 11: Ablation study for multi-granularity masking strategy in explicit procedure modeling stage on Spot-the-Diff.

| Settings | B ↑ | M ↑ | R ↑ | C ↑ |
|---|---|---|---|---|
| w/o Entire Masking | 8.8 | 11.9 | 30.2 | 32.5 |
| w/o Random Patch Masking | 10.3 | 12.5 | 32.7 | 40.7 |
| w/o In-block Masking | 7.9 | 12.0 | 28.0 | 30.0 |
| w/o Out-of-block Masking | 8.0 | 12.1 | 27.6 | 30.5 |
| w/ All Masking Strategies | **11.0** | **13.6** | **33.7** | **42.7** |

**Impact of multi-granularity masking strategy.**    Table 11 shows the contribution of each masking strategy described in Sec 3.1.3. Without the entire masking strategy, the model cannot adequately learn to reconstruct intermediate frames solely from change captions, thereby weakening its cross-modal understanding ability. In contrast, incorporating random patch masking yields better performance by promoting the learning of distributed visual representations. Furthermore, the

significant performance drop observed when either in-block or out-of-block masking is removed highlights the crucial role of these strategies in facilitating spatial-temporal understanding. Figure 7 illustrates the performance comparison across different probability configurations of the four masking strategies. Together with Table 11, the observations consistently reveal three key findings: (1) Stronger learning of distributed visual representations leads to better performance. Random patch masking plays a central role by providing broad and dense visual coverage, and therefore receives the highest probability. (2) Entire masking, in-block masking, and out-of-block masking are essential for modeling global context and localized structural cues. However, overemphasizing any of these structured strategies removes too many fine-grained visual details, which hampers detailed feature learning and ultimately degrades change-detection performance. This is evident from the steady performance drop observed when the probability of any of these three strategies is increased. (3) The four masking strategies work synergistically, jointly supporting both coarse-grained and fine-grained representation learning. In contrast, relying solely on random patch masking yields only marginal improvements.

**Impact of the procedure encoder's depth.** We investigate the impact of the procedure encoder's depth on the CLEVR-Change and Spot-the-Diff datasets, with results presented in Tables 12 and 13. The results reveal that the optimal encoder depth is dataset-dependent. On CLEVR-Change, performance consistently improves with a deeper encoder, peaking with a 12-layer architecture. This suggests that modeling the changes in CLEVR-Change benefits from a higher-capacity encoder. In contrast, a shallower 4-layer encoder is optimal for Spot-the-Diff, as an overfitting is observed with deeper encoders.

Table 12: Ablation results of using different procedure encoder layers on CLEVR-Change.

| Layers | B | M | R | C |
|---|---|---|---|---|
| 2 | 52.6 | 38.9 | 71.8 | 117.4 |
| 4 | 53.9 | 39.9 | 73.1 | 124.3 |
| 8 | 54.2 | 41.0 | 73.9 | 133.2 |
| 12 | **56.7** | **41.7** | **74.7** | **135.6** |

Table 13: Ablation results of using different procedure encoder layers on Spot-the-Diff.

| Layers | B | M | R | C |
|---|---|---|---|---|
| 2 | 7.4 | 13.0 | 28.4 | 30.2 |
| 4 | **11.0** | **13.6** | **33.7** | **42.7** |
| 8 | 9.4 | 12.0 | 32.1 | 42.2 |
| 12 | 7.4 | 13.5 | 27.8 | 30.2 |

## L ABLATION ON IMPLICIT PROCEDURE CAPTIONING

We further evaluate the contributions of two key components within the implicit procedure captioning on the CLEVR-Change dataset and the Spot-the-Diff dataset.

**Explicit and implicit procedure captioning.** We compare our proposed Implicit Procedure Captioning (using learnable queries) against a baseline that performs Explicit Procedure Captioning (directly encoding synthesized frames). Table 14 shows that our implicit approach with procedure queries achieves superior performance on the CLEVR-Change dataset. The explicit baseline, which relies on synthesized frames, not only incurs higher computational costs but also suffers in performance. We attribute the lower accuracy of explicit procedure modeling to the redundant and noisy temporal information in the generated frames. In contrast, our learnable queries provide a more robust representation of procedural dynamics, leading to more accurate change descriptions.

Table 14: Impact of implicit procedure captioning using procedure queries. The first line denotes explicit procedure captioning using synthetic pseudo-frames generated from Procedure Generation Module directly.

| Settings | TPS | B | M | R | C |
|---|---|---|---|---|---|
| Explicit procedure captioning | 421.03 | 56.5 | 40.8 | 74.4 | 128.5 |
| Implicit procedure captioning | 699.04 | 56.7 | 41.7 | 74.7 | 135.6 |

**Impact of the text decoder's depth.** We analyze the effect of decoder depth on the CLEVR-Change and Spot-the-Diff datasets (Tables 15 and 16), observing a general trend of overfitting with excessive layers. The optimal decoder depth for Spot-the-Diff (3 layers) is greater than for CLEVR-Change (2 layers). We attribute this to the nature of the target change descriptions. Unlike the highly structured descriptions for CLEVR-Change, Spot-the-Diff requires more descriptive power. Its surveillance-style scenes feature non-canonical object poses and complex background clutter, demanding greater linguistic capacity from decoder.

Table 15: Ablation results of using different text decoder layers on CLEVR-Change.

| Layers | B | M | R | C |
|---|---|---|---|---|
| 2 | 56.7 | **41.7** | **74.7** | **135.6** |
| 3 | 56.7 | 41.4 | **74.7** | 129.5 |
| 4 | 56.5 | 40.7 | 74.6 | 129.7 |
| 5 | **56.8** | 41.0 | **74.7** | 130.4 |

Table 16: Ablation results of using different text decoder layers on Spot-the-Diff.

| Layers | B | M | R | C |
|---|---|---|---|---|
| 2 | 9.4 | 12.0 | 32.6 | 37.1 |
| 3 | **11.0** | **13.6** | **33.7** | **42.7** |
| 4 | 7.1 | 10.7 | 28.1 | 31.7 |
| 5 | 8.1 | 11.7 | 27.5 | 28.5 |

# M  QUALITATIVE RESULTS

## M.1  COMPARISON OF CAPTIONING GENERATIONS

Figure 8 presents the qualitative results of our ProCap. We compare our model with two non-LLM-based approaches (DIRL (Tu et al., 2024a) and SCORER (Tu et al., 2023c)) and one LLM-based method (FINER (Zhang et al., 2024)) to highlight its generation capabilities. Our model demonstrates robust performance across a variety of change scenarios. Moreover, by incorporating temporal information into the change captioning process, our model better captures the temporal order of events, enabling it to generate more accurate and coherent captions, as exemplified in Figure 8 (j).

## M.2  VISUALIZATION OF CHANGE PROCEDURES

Figures 9-12 present qualitative visualizations of the explicit change procedures generated by our model on three datasets: CLEVR-Change, Spot-the-Diff, and Image-Editing-Request. Our model leverages the synthetic procedures from the Procedure Generation Module and the key frames selected by the Confidence-based Frame Sampling Module to effectively capture the transformation process between image pairs. Notably, it remains robust even when the synthesized procedures exhibit temporal redundancy in the third and fourth samples, which is a critical prerequisite for the subsequent Implicit Procedure Captioning.

## M.3  CASES WITH SIGNIFICANT VIEWPOINT SHIFT

Figure 13 shows several cases exhibiting significant viewpoint shifts in the CLEVR-Change dataset. Following Park et al. (2019), we use the IoU between similar image pairs to quantify the degree of viewpoint change. The mean IoU in CLEVR-Change is 0.51 with a variance of 0.02; therefore, an IoU around 0.2 is regarded as indicating a substantial viewpoint shift (as illustrated in the first two rows). Notably, even under such drastic viewpoint differences, our model is able to reconstruct a plausible intermediate process, demonstrating the robustness of our procedure modeling module. We attribute this robustness to our proposed consistency loss, which explicitly promotes spatial-temporal consistency in the reconstructed intermediate frames.

## M.4  FAILURE CASES

Figure 14 presents several failure cases produced by our proposed ProCap. For most failures on the CLEVR-Change dataset, the modifications are extremely subtle, which makes it difficult for the model to reliably detect the change throughout the procedure. In contrast, the primary source of errors in the Image-Editing-Request and Spot-the-Diff datasets lies in inaccurate reconstruction of the intermediate procedure, which subsequently leads to incorrect change captions. We attribute this issue to overfitting, as these two datasets are more open and unconstrained compared with the

**(a) Color Change**
before          after

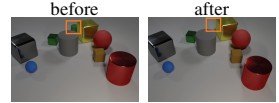

**Ground Truth:** the tiny yellow cube turned gray
**ProCap (Ours):** the small yellow rubber cube that is behind the big yellow matte object changed to gray
**DIRL:** the scene remains the same
**SCORER:** the scene remains the same
**FINER:** there is no change

**(d) Object Remove**
before          after

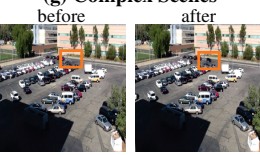

**Ground Truth:** the tiny green cylinder is no longer there
**ProCap (Ours):** the small green matte cylinder that is behind the big gray matte cylinder is no longer there
**DIRL:** the scene remains the same
**SCORER:** the small yellow rubber cube that is behind the small brown rubber block became red
**FINER:** there is no change

**(g) Complex Scenes**
before          after

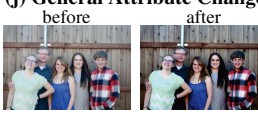

**Ground Truth:** a black suv has appeared
**ProCap (Ours):** a black vehicle is driving down the road in the second image
**DIRL:** there is a person walking in the parking lot
**FINER:** there is a person walking in the parking lot

**(j) General Attribute Change**
before          after

**(b) Texture Change**
before          after

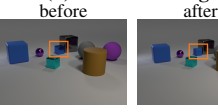

**Ground Truth:** the small blue shiny cube behind the large gray rubber object became rubber
**ProCap (Ours):** the small blue metal cube that is behind the large gray matte ball changed to rubber
**DIRL:** the scene remains the same
**SCORER:** the scene is the same as before
**FINER:** there is no change

**(e) Object Move**
before          after

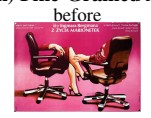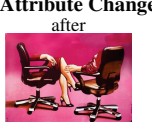

**Ground Truth:** the shiny cylinder changed its location
**ProCap (Ours):** the small green metal cylinder that is in front of the small red matte object changed its location
**DIRL:** the scene remains the same
**SCORER:** the scene is the same as before
**FINER:** there is no change

**(h) Fine-Grained Attribute Change**
before          after

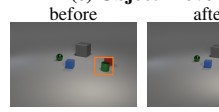

**Ground Truth:** Remove the text from the entire image
**ProCap (Ours):** remove the text from the photo
**DIRL:** Change the snake on the snake
**FINER:** change the background to pink

**(k) General Attribute Change**
before          after

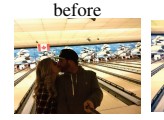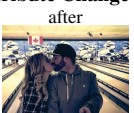

**Ground Truth:** crop photo
**ProCap (Ours):** crop the image
**DIRL:** Change the background from that it looks like a little
**FINER:** remove the people in the background

**(c) Object Add**
before          after

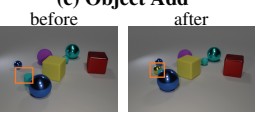

**Ground Truth:** the yellow ball has been newly placed
**ProCap (Ours):** the small yellow metal ball that is in front of the big blue object has been newly placed
**DIRL:** the small cyan matte ball that is in front of the big red metal thing became yellow
**FINER:** the small cyan matte ball that is in front of the big red cube changed to metallic

**(f) Complex Scenes**
before          after

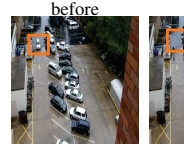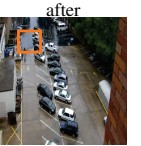

**Ground Truth:** the white vehicle driving in the lot on the left is gone
**ProCap (Ours):** a silver car has left the parking lot
**DIRL:** there is a person walking in the parking lot
**FINER:** there is a person walking in the parking lot

**(i) Composite Change**
before          after

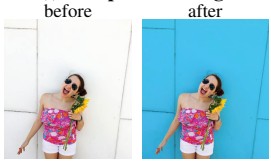

**Ground Truth:** Change the wall color from white to blue
**ProCap (Ours):** change the color of the image to a blue
**DIRL:** change the background from a little bit
**FINER:** zoom in on the girl

**(l) Pattern Change**
before          after

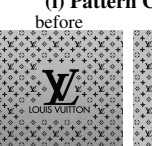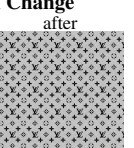

**Ground Truth:** remove the text "louis vuitton" and the "LV" sign, add the same area that is behind and above instead of the text and the sign
**ProCap (Ours):** remove the text
**DIRL:** remove all background
**FINER:** remove the background

**Ground Truth:** Darken the entire image
**ProCap (Ours):** darken the image
**DIRL:** Brighten the image
**FINER:** brighten the entire image

Figure 8: Comparison of captioning generations. We compare our model against two non-LLM-based approaches (DIRL and SCORER) and one LLM-based method (FINER). The examples are grouped into 10 change types, and (a)-(e) are from the CLEVR-Change dataset, (f)-(g) from Spot-the-Diff, and (h)-(l) from Image-Editing-Request.

CLEVR-Change dataset. In future work, we plan to further investigate the generation and modeling of more coherent and semantically reasonable intermediate transformation processes to improve the robustness of change captioning.

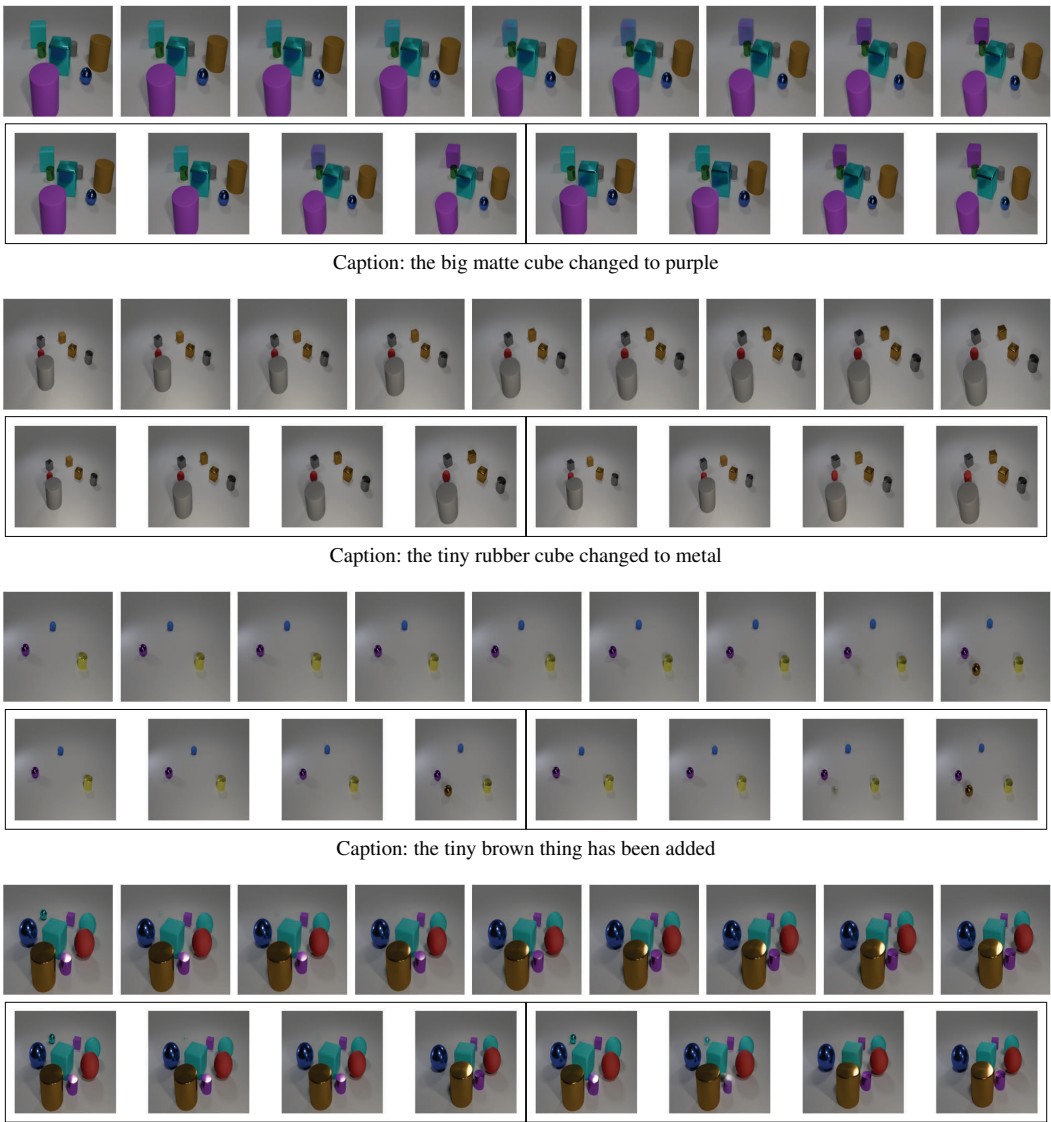

Figure 9: Visualization of change procedures on CLEVR-Change. For each sample, the **top** row displays the synthetic procedure generated by the Procedure Generation Module. The **bottom-left** shows key frames selected from this synthetic procedure using the Confidence-based Frame Sampling Module, while the **bottom-right** visualizes the reconstructed procedural representation produced by the Procedure Encoder within the Procedure Modeling Module.

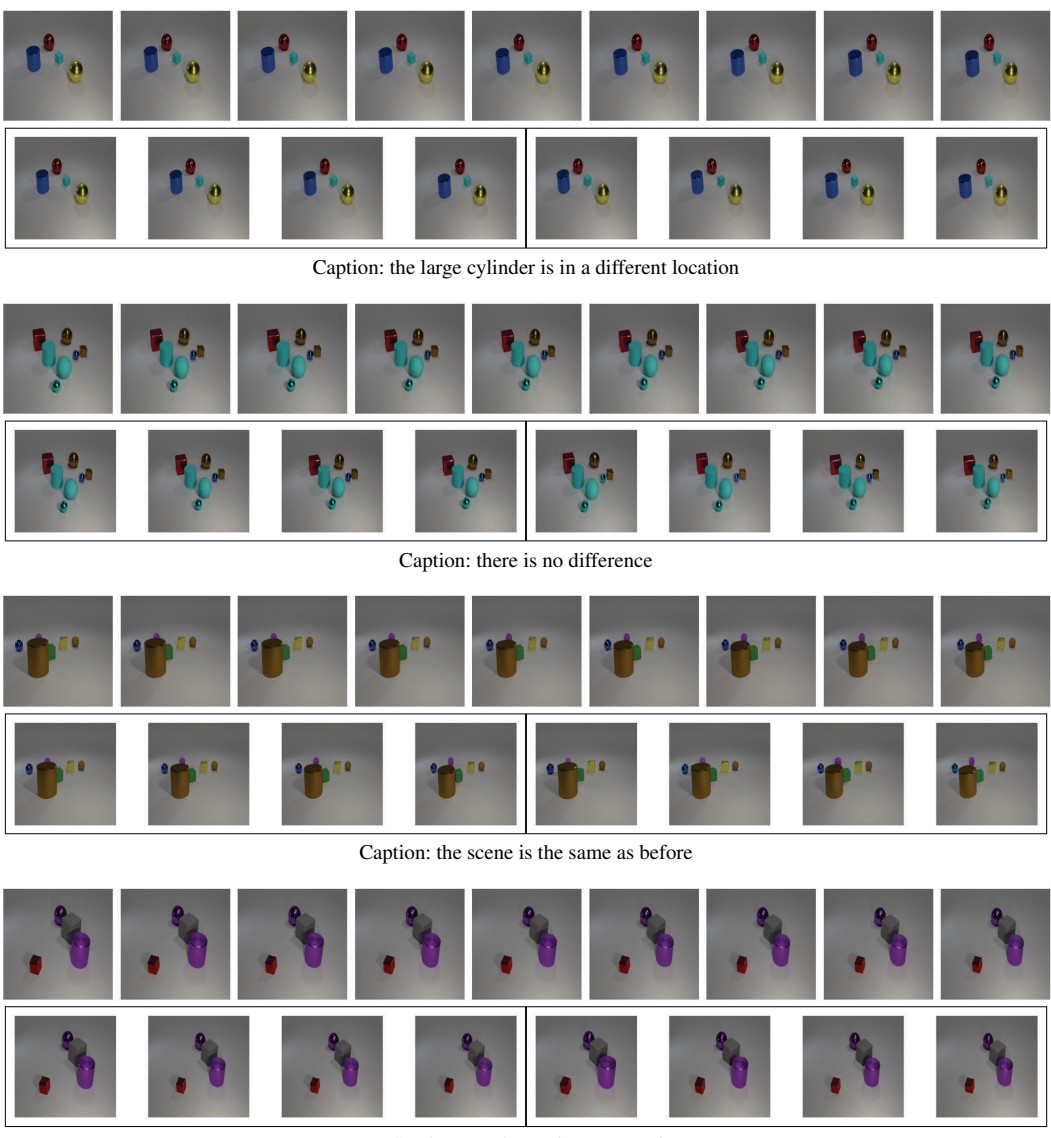

Figure 10: Additional visualizations of change procedures on CLEVR-Change.

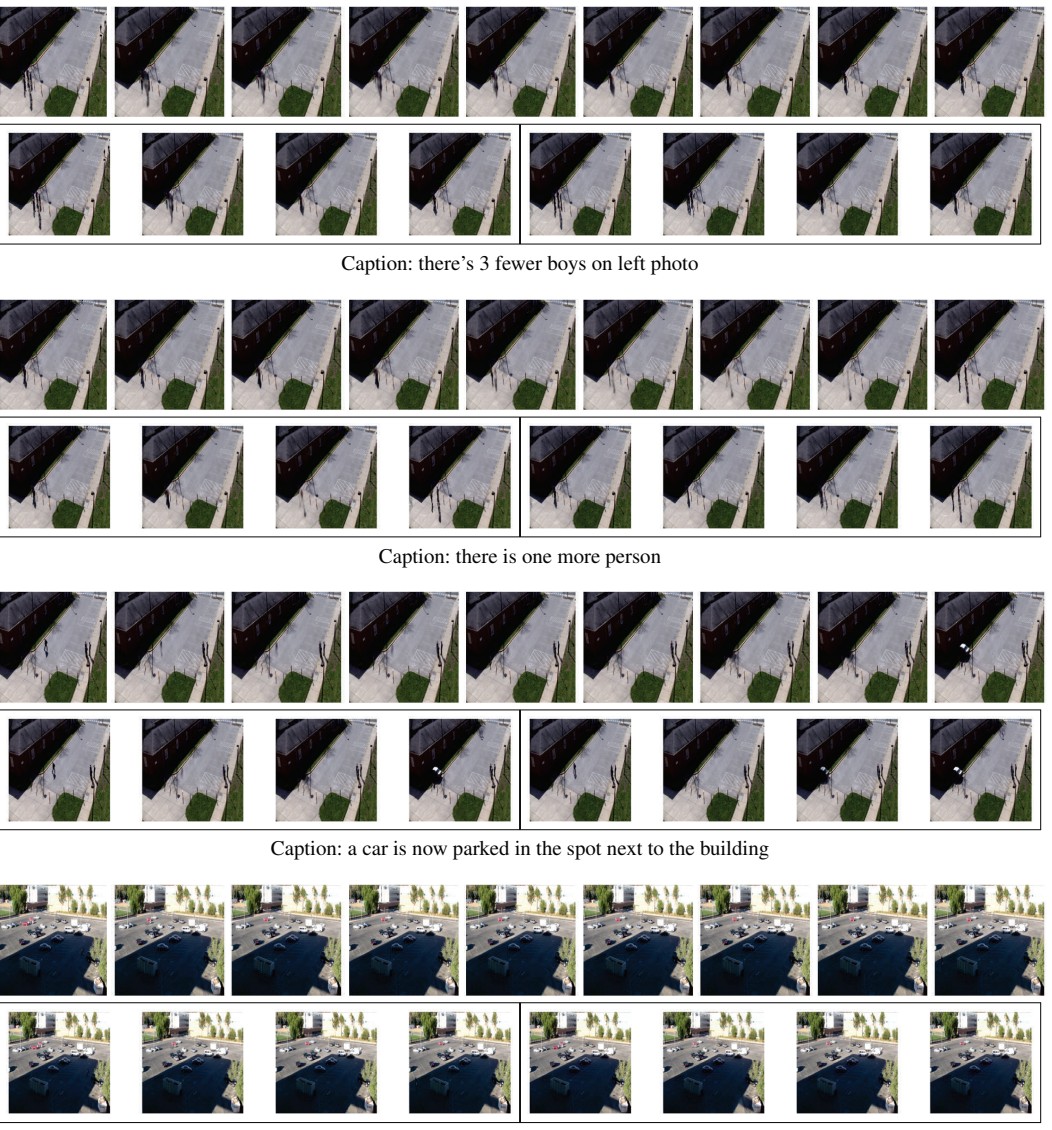

Figure 11: Visualization of change procedures on Spot-the-Diff.

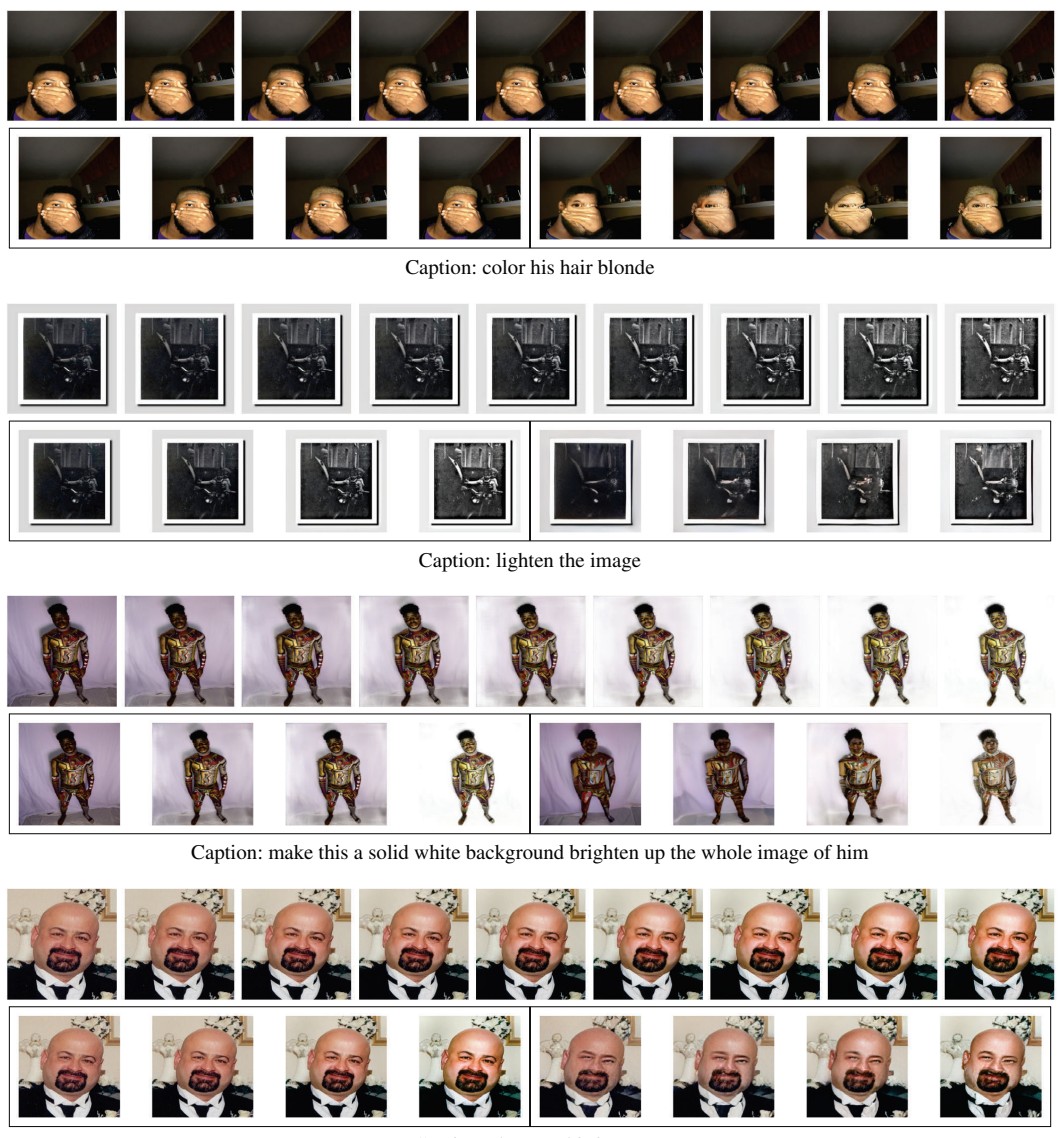

Caption: color his hair blonde

Caption: lighten the image

Caption: make this a solid white background brighten up the whole image of him

Caption: sharpen this image

Figure 12: Visualization of change procedures on Image-Editing-Request.

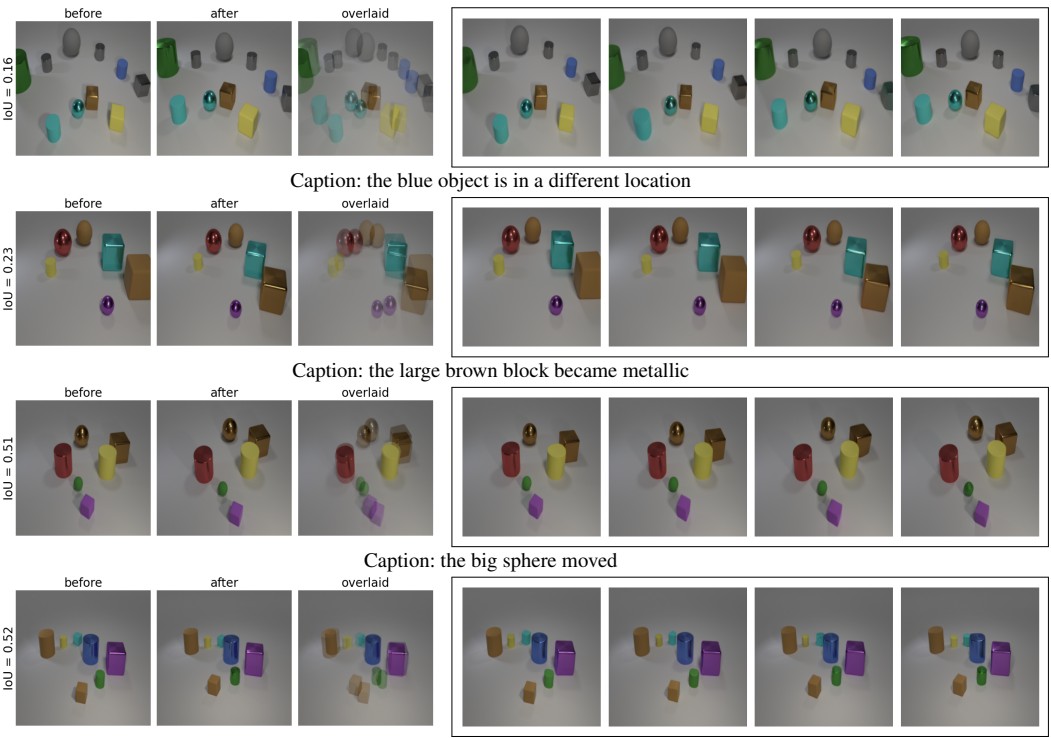

Figure 13: Visualization of cases with significant viewpoint shift. The **left** shows the original image pair with the overlaid image. The **right** visualizes the reconstructed procedural representation produced by the Procedure Encoder within the Procedure Modeling Module.

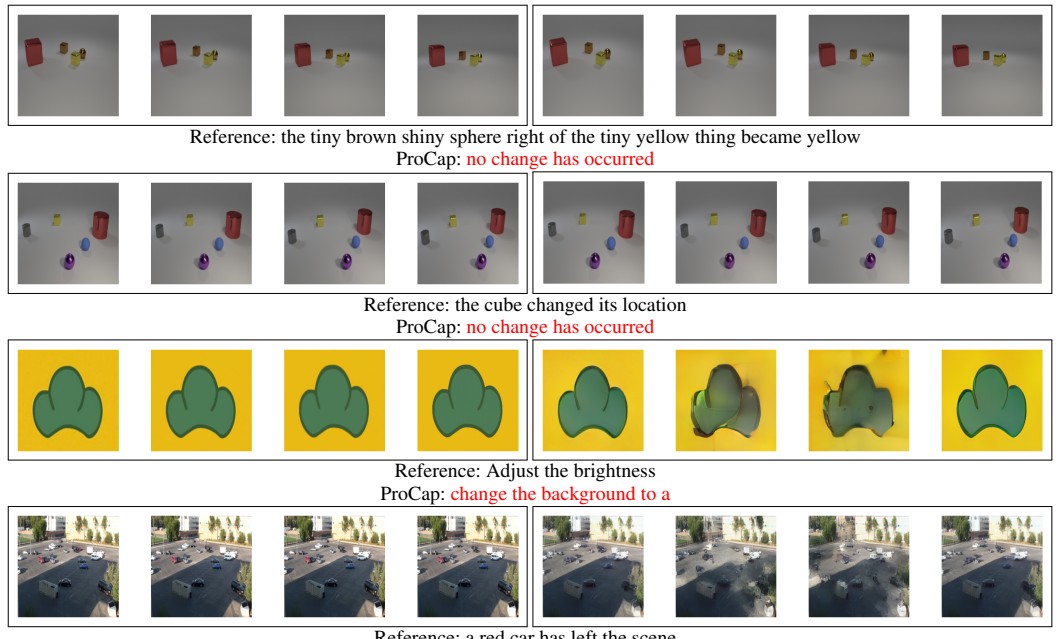

Figure 14: Visualization of failure cases generated by ProCap. The **left** shows key frames selected from the synthetic procedure using the Confidence-based Frame Sampling Module, while the **right** visualizes the reconstructed procedural representation produced by the Procedure Encoder within the Procedure Modeling Module.

## N  LIMITATION AND FUTURE WORK

In this work, we propose a novel two-stage framework, ProCap, which reformulates change captioning from static comparison to dynamic procedure modeling. While experiments demonstrate that our method achieves strong performance across three widely-used benchmark datasets, certain challenges remain in specific scenarios.

For instance, when scenes exhibit dramatic changes, for example, where transformations exceed the variations in position, appearance, and existence defined in Sec. 3, or drastic viewpoint changes happen, generating perfectly physically grounded intermediate frames becomes inherently ill-posed for any current generative model, as pixel-level correspondence is no longer preserved. In such cases, 2D generative models, such as optical-flow-based approaches (Lu et al., 2022), face fundamental limitations due to the lack of explicit geometric depth reasoning. We believe that a paradigm shift toward 3D scene modeling to maintain geometric consistency is beneficial to maintain geometric consistency and produce physically grounded intermediate frames under such extreme variations. Consequently, we identify 3D-aware representation as a critical direction to extreme geometric discontinuities in future exploration.

Another open problem lies in defining what constitutes a theoretically optimal informative point. While our current formulation provides a practical solution, a more rigorous theoretical definition remains unexplored. Future work could investigate a principled mathematical characterization of this optimal point within the broader context of change analysis, potentially leading to more robust and generalizable criteria.

Finally, integrating LLMs represents a natural and valuable extension of our framework. We plan to explore LLM-based architectures—such as instruction-tuning strategies—to combine the high-level reasoning capability of LLMs with the explicit dynamic modeling strengths of ProCap. Such integration may enable richer semantic guidance and more adaptive dynamic understanding in future systems.

Collectively, we believe these limitations highlight several promising avenues for continued development. With more refined model design and deeper theoretical grounding, ProCap can be extended to address these challenges more effectively.

## O  ETHICS STATEMENT

This work adheres to the ICLR Code of Ethics. No human subjects or animal experiments were involved in this study. All datasets used, including CLEVR-Change, Spot-the-Diff, and Image-Editing-Request, were obtained in accordance with their respective usage guidelines, ensuring full compliance with privacy standards. We have taken care to minimize potential biases and avoid discriminatory outcomes throughout the research process. No personally identifiable information was utilized, and no experiments were conducted that could raise privacy or security concerns. We are committed to upholding transparency, fairness, and integrity in all aspects of this research.

## P  REPRODUCIBILITY STATEMENT

We have taken extensive measures to ensure the reproducibility of our results. All code and data used in the experiments will be released publicly to facilitate replication and independent verification. The experimental setup—including training procedures, model configurations, and hardware specifications—is detailed in Appendix H. In addition, we provide a comprehensive description of ProCap to further support reproducibility.

Furthermore, the three change captioning datasets used in our work—CLEVR-Change, Spot-the-Diff, and Image-Editing-Request—are publicly available, ensuring consistent and reproducible evaluation.

We believe these efforts will enable other researchers to faithfully reproduce our findings and contribute to advancing the field.

## Q  STATEMENT OF USING LLMS IN THE PAPER

Large Language Models (LLMs) were employed to assist in writing and refining this manuscript, specifically for grammar checking and sentence polishing, with the aim of enhancing overall readability.

Importantly, the LLM was not involved in the ideation, research methodology, experimental design, or data analysis. All research concepts, ideas, and analyses were independently developed and carried out by the authors. The role of the LLM was strictly limited to improving the linguistic quality of the text, without contributing to the scientific content.

The authors take full responsibility for the manuscript, including any portions refined with LLM assistance. We have ensured that the use of LLMs complies with ethical standards and does not involve plagiarism or scientific misconduct.

