# OpenReview forum: "Imagine How To Change: Explicit Procedure Modeling for Change Captioning"
_ICLR.cc/2026/Conference — ICLR 2026 Poster_

### Official Review · Reviewer_hLpn · 2025-10-28

**Soundness:** 3
**Presentation:** 3
**Contribution:** 3
**Rating:** 6
**Confidence:** 5

**Summary:**

The paper presents a novel perspective on the task of Change Captioning. Instead of merely focusing on static image pairs, it emphasizes modeling the changing process between similar image pairs. This process captures rich spatial-temporal information, enabling the model to better understand how changes occur and to generate more precise descriptions of them. Extensive experiments demonstrate that, even without a complex framework design, the proposed method effectively leverages the learning of dynamic change processes to overcome challenges caused by distractor variations, achieving strong and robust performance.

**Strengths:**

1. The paper provides a fresh and meaningful perspective on the change captioning task by modeling the dynamic process of change instead of focusing on static image pairs, which makes the approach both novel and insightful.
2. The proposed framework, ProCap, demonstrates an elegant design that effectively captures rich spatial–temporal information by explicitly modeling the changing process. Moreover, its use of implicit procedure queries enables efficient and effective inference. Experiments show that ProCap surpasses existing non-LLM-based methods on two datasets and achieves competitive results on the remaining one.

**Weaknesses:**

1.As shown in Table 1, the method performs slightly worse on Spot-the-Diff; further analysis would be informative.
2.Presenting some failure cases would provide deeper insights into its limitations.

**Questions:**

Please refer to weaknesses.

---

> ### Author Response · Authors · 2025-11-25
> **Response to Reviewer hLpn**
>
> We appreciate the reviewer for the valuable comments, which have greatly helped us improve the quality of our manuscripts.
>
> ---
> **W1:** The performance of the proposed method is slightly worse on the Spot-the-Diff dataset.
>
> **A:** We thank the reviewer for the comment. We would like to clarify that the performance on Spot-the-Diff is highly sensitive to the richness of the training captions. **The competing method MCT-CCDiff [1] expands the Spot-the-Diff training set using GPT-generated captions, while our default setting uses only the original annotations.** This leads to an uneven comparison, particularly on linguistically sensitive metrics such as METEOR and ROUGE.
>
> To isolate the effect of data rather than model design, we conduct an additional experiment where we adopt a similar data-augmentation strategy using Qwen3 [2] to enrich the Spot-the-Diff training captions as shown in the following table. Under this matched setting, our model achieves competitive M and R scores and surpasses MCT-CCDiff on the more semantically aligned metrics, especially CIDEr and BLEU-4 (CIDEr: 44.55 vs. 41.7, BLEU-4: 11.7 vs. 10.8).
>
> | Method                       | B $\uparrow$   | M $\uparrow$   | R $\uparrow$   | C $\uparrow$   |
> |------------------------------|------|------|------|------|
> | MCT-CCDiff [1]               | 10.8 | 14.5 | 35.5 | 41.7 |
> | Ours (w/o data augmentation) | 11.0 | 13.6 | 33.7 | 42.7 |
> | Ours (w/ data augmentation)  | 11.7 | 14.2 | 34.6 | 44.6 |
>
>
> These results indicate that the previously observed gap on Spot-the-Diff is primarily due to **differences in training data** rather than the robustness of our model. When trained under comparable conditions, ProCap performs on par with or better than prior SOTA methods, further validating the effectiveness and robustness of our approach. We have included this discussion on the Spot-the-Diff dataset in the revised version.
>
> [1] Hu, Jinhong, et al. "MCT-CCDiff: Context-aware Contrastive Diffusion model with Mediator-bridging Cross-modal Transformer for Image Change Captioning." IEEE Transactions on Image Processing (2025).
>
> [2] Yang, An, et al. "Qwen3 technical report." arXiv preprint arXiv:2505.09388 (2025).
>
> ---
> **W2:** Presenting some failure cases would provide deeper insights into the model’s limitations.
>
> **A:** We appreciate the reviewer’s suggestion. We agree that failure cases can provide valuable insight into the model’s limitations. In the revised version, we include representative failure examples (such as cases involving extremely subtle changes, inaccurate procedure generation, and low-resolution inputs) along with a brief analysis of the underlying causes.

---

> > ### Comment · Reviewer_hLpn · 2025-11-27
> >
> > After reading the authors' response to another reviewer, I would be very interested in further clarification on an underlying assumption: how does the model benefit from learning a continuous visual trajectory γ_T, and why is this intermediate representation expected to be easier for a small model than directly learning the caption? This seems central to the contribution of "procedure modeling". A more detailed explanation would significantly strengthen the paper.

---

> > > ### Author Response · Authors · 2025-11-27
> > >
> > > We thank the reviewer for the interest and the suggestion. We clarify that the goal of change captioning is not simply to generate text, but to first *identify and understand* the visual differences between two images and then describe them. Directly mapping an image pair to a caption forces a small model to perform both fine-grained change localization and language generation in a single step, which is highly challenging, especially when the visual differences are subtle.
> > >
> > > The procedure modeling we proposed is designed precisely to decompose this difficult problem. As discussed in Sec. 3, the procedure models *how the scene evolves* from the “before’’ state to the “after’’ state. This provides the model with an explicit, structured intermediate representation of the change dynamics, i.e., **not only what has changed but how it occurs**. This decomposition significantly reduces the reasoning burden for a small model: instead of inferring changes implicitly from two static frames, the model receives a smoother, temporally coherent transition path that is easier to interpret and convert into language.
> > >
> > > We acknowledge that the unconstrained mapping $\gamma_T$ can yield infinitely many procedures, making it difficult for a small model to learn meaningful dynamics. To address this, we impose a strong physical constraint using an optical-flow-based FI model, which largely restricts the solution space. As shown in Table 9, replacing this with a weaker constraint (e.g., diffusion-based interpolation) significantly degrades performance, confirming the importance of the strong constraint. Moreover, the performance gap between k=0 and k=1 in Tables 2 and 7 indicates that small models struggle to reason directly from two static frames but benefit substantially from our procedure modeling. Together, these results demonstrate that the modeled procedure provides a structured and learnable intermediate representation that simplifies change reasoning and enables a small model to generate more accurate and coherent captions.

---

> > > > ### Comment · Reviewer_hLpn · 2025-11-27
> > > >
> > > > Thanks for the reponse and my concerns are sovled. I have raised the score.

---

### Official Review · Reviewer_weBZ · 2025-10-31

**Soundness:** 3
**Presentation:** 3
**Contribution:** 3
**Rating:** 6
**Confidence:** 5

**Summary:**

This paper addresses the task of Change Captioning, which aims to generate textual descriptions of the differences between two similar images. The authors argue that existing methods, which rely on static image pair comparison, fundamentally fail to model the rich temporal dynamics of how a change occurs. To address this limitation, the paper proposes ProCap, a novel two-stage framework that reformulates the task from static comparison to dynamic procedure modeling.
Stage 1: Explicit Procedure Modeling: This stage trains a "procedure encoder" to capture the latent dynamics of a change.
Stage 2: Implicit Procedure Captioning: This stage handles the final captioning task.

**Strengths:**

1.	Novel Problem Formulation: The paper introduces a conceptual shift by reformulating change captioning from a static image comparison task into a dynamic procedure modeling problem. This directly addresses a key limitation of prior work, which largely ignores the rich temporal dynamics of how a change unfolds.
2.	Efficient Architecture:
(1)	Stage 1 effectively learns a rich representation of spatio-temporal dynamics by training an encoder on explicitly generated and sampled keyframes.
(2)	Stage 2 introduces learnable procedure queries as an efficient solution. This allows the model to implicitly reason about the change process during inference, bypassing the computational cost and noise associated with synthesizing frames.

**Weaknesses:**

1.	It does not show a clear performance advantage on Spot-the-Diff over other SOTA methods.
2.	As shown in Table 4, the consistency loss does not yield a substantial performance gain.

**Questions:**

1.	When there are significant scene differences between the two images, how does the procedure modeling module generate intermediate frames that are meaningful for the change analysis?
2.	The ablation studies were performed exclusively on the CLEVR-Change dataset, which is insufficient for evaluating the proposed method's effectiveness in real-world scenarios (such as Spot-the-Diff).

---

> ### Author Response · Authors · 2025-11-24
> **Response to Reviewer weBZ (Part 1)**
>
> We thank the reviewer for the thoughtful feedback.
>
> ---
> **W2:** The consistency loss does not yield a substantial performance gain, as shown in Table 4.
>
> **A:** We thank the reviewer for the comment. We clarify that although the consistency loss alone does not bring a large performance gain, it **plays a structurally necessary and complementary role that cannot be replaced by the alignment loss**. The purpose of the consistency loss is to **ensure the spatial-temporal consistency** of the procedure, different from the alignment loss, which focuses on grounding the visual and linguistic representation, contributing directly to the multimodal understanding.
>
>
> As can be observed from Table 4, using the alignment loss alone fails to reach the best performance because the model lacks an explicit mechanism to enforce step-wise temporal coherence. Such an effect can also be observed in the real-world Spot-the-Diff dataset, as shown in the following table. When the alignment loss is present, adding the consistency loss consistently leads to improvements, indicating that the consistency loss is not designed to provide large standalone gains, but to enable the alignment loss to function reliably by **stabilizing temporal logic**.
>
> | $\mathcal{L}_\text{msm}$ | $\mathcal{L}_\text{align}$ | $\mathcal{L}_\text{csy}$ | B $\uparrow$  | M $\uparrow$  | R $\uparrow$  | C $\uparrow$  |
> |-------|---------|--------|-----|-----|------|------|
> | $\checkmark$     |         |        | 8.1 | 11.8 | 28.1 | 29.7 |
> | $\checkmark$     |         | $\checkmark$      | 7.9 | 11.7 | 28.0 | 28.9 |
> | $\checkmark$     | $\checkmark$       |        | 9.3 | 12.5 | 31.2 | 36.3 |
> | $\checkmark$     | $\checkmark$       | $\checkmark$      | **11.0** | **13.6** | **33.7** | **42.7** |
>
> ---
> **Q1:** How does the procedure modeling module generate meaningful intermediate frames for change analysis when significant scene differences happen between the two images?
>
> **A:** We thank the reviewer for the insightful question. We use the IoU between image pairs in the CLEVR-Change dataset to quantify the degree of viewpoint shift and analyze how well our procedure modeling module handles such challenging cases. In the revised manuscript (Appendix L, Figure 12), we provide representative examples with IoU $\approx$ 0.2, a level commonly regarded in prior work [1] as indicating substantial viewpoint differences. These cases show that our procedure modeling module produces semantically coherent and temporally meaningful intermediate frames, demonstrating its robustness under significant viewpoint changes. This robustness arises from our consistency loss, which explicitly promotes spatial-temporal consistency in the reconstructed intermediate frames.
>
> We acknowledge that generating meaningful procedures under extreme scene differences (e.g., collapsed pixel-level correspondence and completely disjoint views) remains an open challenge. In such cases, 2D generative models, such as optical-flow-based approaches [2], face fundamental limitations due to the lack of explicit geometric depth reasoning. We believe that a paradigm shift toward 3D scene modeling to maintain geometric consistency is beneficial to maintain geometric consistency and produce physically grounded intermediate frames under such extreme variations. Consequently, we identify **3D-aware representation** as a critical direction to extreme geometric discontinuities in future exploration. We have included this discussion in the Future Work section of our revised manuscript.
>
> [1] Park, Dong Huk, Trevor Darrell, and Anna Rohrbach. "Robust change captioning." Proceedings of the IEEE/CVF International Conference on Computer Vision. 2019.
>
> [2] Lu, Liying, et al. "Video frame interpolation with transformer." Proceedings of the IEEE/CVF Conference on Computer Vision and Pattern Recognition. 2022.

---

> ### Author Response · Authors · 2025-11-25
> **Response to Reviewer weBZ (Part 2)**
>
> **Q2:** It is insufficient to evaluate the proposed method’s effectiveness in real-world scenarios with only ablation studies on the CLEVR-Change dataset.
>
> **A:** We thank the reviewer for pointing this out. We agree that examining the method under realistic conditions is important. We have included the ablation study to the Spot-the-Diff dataset in the revised version to comprehensively demonstrate the method’s effectiveness in real-world scenarios. The ablation studies on the Spot-the-Diff dataset are listed below, which demonstrate our proposed method’s generalization in real-world scenarios.
>
> | Explicit procedure modeling (Stage 1) | Implicit procedure captioning (Stage 2) | B $\uparrow$        | M $\uparrow$        | R $\uparrow$        | C $\uparrow$        |
> | :---: | :---: | --- | --- | --- | --- |
> |  |  | 7.9 | 11.7 | 28.0 | 28.9 |
> | $\checkmark$ |  | 8.5 | 12.1 | 27.8 | 30.6 |
> | | $\checkmark$ | 8.3 | 12.1 | 27.5 | 29.8 |
> | $\checkmark$ | $\checkmark$ | 8.6 | 12.5 | 32.2 | 36.0 |
>
> | Frame num $k$ | B $\uparrow$        | M $\uparrow$        | R $\uparrow$        | C $\uparrow$        |
> |:-------------:|-----------|-----------|-----------|-----------|
> | 1           | 8.6 | 12.5 | 32.2 | 36.0 |
> | 2           | **11.0**  | **13.6**  | **33.7**  | **42.7**  |
> | 4           | 8.5       | 12.4      | 27.7      | 31.3      |
> | 7           | 7.5       | 11.8      | 25.7      | 29.2      |
>
> | $\mathcal{L}_\text{msm}$ | $\mathcal{L}_\text{align}$ | $\mathcal{L}_\text{csy}$ | B $\uparrow$  | M $\uparrow$  | R $\uparrow$  | C $\uparrow$  |
> |-------|---------|--------|-----|-----|------|------|
> |  $\checkmark$     |         |        | 8.1 | 11.8 | 28.1 | 29.7 |
> | $\checkmark$     |         | $\checkmark$      | 7.9 | 11.7 | 28.0 | 28.9 |
> | $\checkmark$     | $\checkmark$       |        | 9.3 | 12.5 | 31.2 | 36.3 |
> | $\checkmark$    | $\checkmark$       | $\checkmark$      | **11.0** | **13.6** | **33.7** | **42.7** |

---

> ### Author Response · Authors · 2025-11-25
> **Response to Reviewer weBZ (Part 3)**
>
> **W1:** It does not show a clear performance advantage on Spot-the-Diff over other SOTA methods.
>
> **A:** We thank the reviewer for the comment. We would like to clarify that the Spot-the-Diff dataset behaves differently from the other two benchmarks and is known to be highly sensitive to the linguistic richness of its training captions. **As reported in MCT-CCDiff [1], the method substantially expands the Spot-the-Diff training set using GPT-generated captions, while our default setting uses only the original annotations.** This leads to differences mainly on linguistically sensitive metrics (e.g., METEOR, ROUGE), which reflect caption surface forms rather than the underlying modeling capability.
>
> To assess the method’s robustness under comparable conditions, we conduct an additional experiment using the same augmentation strategy as MCT-CCDiff, where we expand the Spot-the-Diff training captions with Qwen3 [2]. Under this matched setting, ProCap achieves similar METEOR and ROUGE scores, and **outperforms MCT-CCDiff on the more semantically aligned measures**, including CIDEr and BLEU-4.
>
> | Method                       | Speed $\downarrow$ (s/caption, used in MCT-CCDiff) | B $\uparrow$   | M $\uparrow$   | R $\uparrow$   | C $\uparrow$   |
> |------------------------------|:-------------------------------------------:|:------:|:------:|:------:|:------:|
> | MCT-CCDiff [1]               | 0.91                                      | 10.8 | 14.5 | 35.5 | 41.7 |
> | Ours (w/o data augmentation) | 0.04                                      | 11.0 | 13.6 | 33.7 | 42.7 |
> | Ours (w/ data augmentation)  | 0.04                                      | 11.7 | 14.2 | 34.6 | 44.6 |
>
> Beyond accuracy, our method is also substantially more efficient than existing SOTA methods. As shown in the table below, ProCap is **22× faster** than MCT-CCDiff under identical inference settings. This demonstrates that our approach offers a significantly **better efficiency–effectiveness trade-off**, further highlighting its practical advantages. We have added this discussion in the revised version.
>
> [1] Hu, Jinhong, et al. "MCT-CCDiff: Context-aware Contrastive Diffusion model with Mediator-bridging Cross-modal Transformer for Image Change Captioning." IEEE Transactions on Image Processing (2025).
>
> [2] Yang, An, et al. "Qwen3 technical report." arXiv preprint arXiv:2505.09388 (2025).

---

### Official Review · Reviewer_REM5 · 2025-11-01

**Soundness:** 3
**Presentation:** 2
**Contribution:** 3
**Rating:** 6
**Confidence:** 3

**Summary:**

In this paper, the authors focused on the task of change captioning. Considering the existing methods ignore the temporal dynamics of the change procedure, the authors proposed the ProCap framework. Specifically, ProCap explicitly models the spatio-temporal changes between image pairs. Based on the first stage, the second stage can generate rich descriptions by implicitly reasoning over the modeled change procedure.

**Strengths:**

1. The designed framework reformulates change captioning from static comparison to dynamic procedure modeling, which can capture the rich temporal dynamics.
2. The proposed explicit procedure modeling module can produce continuous frames between static image pairs, which facilitates the change captioning.

**Weaknesses:**

1. The method’s performance across the three datasets does not consistently outperform baseline methods, indicating that its overall robustness may not be sufficiently strong.
2. ProCap employs a non-LLM-based backbone, and it remains unclear how its procedure modeling module would still perform when integrated with a powerful LLM-based backbone.

**Questions:**

As listed above.

---

> ### Author Response · Authors · 2025-11-24
> **Response to Reviewer REM5 (Part 1)**
>
> We thank the reviewer for the detailed review.
>
> ---
> **W2:** It remains unclear how ProCap’s procedure modeling module would perform when integrated with a powerful LLM-based backbone.
>
> **A:** We would like to clarify, the primary motivation of our current design was to rigorously validate that the performance gains stem specifically from our proposed dynamic procedure modeling paradigm. Without relying on the extensive priors of LLMs, we demonstrate that the pure fundamental paradigm shift itself, from static image comparison to dynamic procedure modeling, yields substantial improvements. Consequently, we prioritized the non-LLM setting to strictly evaluate the intrinsic value of procedural modeling against static comparison baselines, aiming to establish a distinct and scientifically rigorous direction for the community.
>
> To evaluate how our procedure modeling module interacts with an LLM-based backbone, we conducted an experiment on the Image-Editing-Request dataset using LLaMA-3.2 [1] as the decoder. The results are shown in the table below. The performance degradation can be attributed primarily to the **feature-dimension mismatch** between our procedure modeling module (768-dim) and the LLM-based backbone (2048-dim).  To adapt our model to an LLM-based backbone, a projection layer is required to transform our encoded vision features to the same dimension as the LLM-based backbone. However, without sufficient large-scale training to properly align this projection, the LLM cannot effectively utilize the procedural information, limiting the overall performance.
>
> We thank the reviewer for the constructive suggestion. We view the integration with LLMs as a natural and important extension of our work. We plan to explore LLM-based architectures (e.g., via instruction tuning strategies) in future work to combine the reasoning power of LLMs with the explicit dynamic modeling of ProCap. We have included this discussion in the Future Work section of our revised manuscript.
>
> | Methods | B $\uparrow$   | M $\uparrow$        | R $\uparrow$        | C $\uparrow$        |
> | --- | --- | --- | --- | --- |
> | Ours (with LLM-backbone) | 10.2 | 14.9 | 41.3 | 38.5 |
> | Ours (non-LLM) | 11.7 | 15.9 | 43.2 | 40.6 |
>
> [1] Grattafiori, Aaron, et al. "The llama 3 herd of models." arXiv preprint arXiv:2407.21783 (2024).

---

> ### Author Response · Authors · 2025-11-25
> **Response to Reviewer REM5 (Part 2)**
>
> **W1:** The method’s performance across the three datasets does not consistently outperform baseline methods, indicating that its overall robustness may not be sufficiently strong.
>
> **A:** We thank the reviewer for the comment. We have observed a performance gap on the Spot-the-Diff dataset compared with the current SOTA method, MCT-CCDiff [1]. However, we attribute this discrepancy primarily to **the differences in the training data captions**, rather than a fundamental limitation in our model's robustness.
>
> As documented in MCT-CCDiff, their training includes a substantial expansion of the original training captions by incorporating GPT-generated captions. In contrast, we utilizes only the original, unaugmented annotations. This difference in training data preparation significantly impacts performance on linguistically sensitive evaluation metrics such as METEOR and ROUGE, which heavily depend on the diversity and richness of training captions.
>
> To evaluate the effect of the dataset captions, we performed an additional experiment on Spot-the-Diff using the same augmentation strategy. We expand the training set with captions generated by Qwen3 [2]. Under this matched setting, our method achieves competitive METEOR and ROUGE scores and outperforms MCT-CCDiff on the more semantically aligned metrics, especially CIDEr and BLEU-4 (+7% improvements on CIDEr, +8% improvements on BLEU-4). This confirms that the previously observed gap is largely **due to training caption differences rather than a lack of robustness in our approach**.
>
> | Method                       | Speed $\downarrow$ (s/caption, used in MCT-CCDiff) | B $\uparrow$   | M $\uparrow$   | R $\uparrow$   | C $\uparrow$   |
> |------------------------------|:-------------------------------------------:|:------:|:------:|:------:|:------:|
> | MCT-CCDiff [1]               | 0.91                                      | 10.8 | 14.5 | 35.5 | 41.7 |
> | Ours (w/o data augmentation) | 0.04                                      | 11.0 | 13.6 | 33.7 | 42.7 |
> | Ours (w/ data augmentation)  | 0.04                                      | 11.7 | 14.2 | 34.6 | 44.6 |
>
>
> Moreover, we further compared inference efficiency. Our method is **22× faster** than MCT-CCDiff under identical conditions while also achieving superior CIDEr performance. This demonstrates that our approach is not only competitive **but both more effective and substantially more efficient** than existing non-LLM SOTA methods.
>
> [1] Hu, Jinhong, et al. "MCT-CCDiff: Context-aware Contrastive Diffusion model with Mediator-bridging Cross-modal Transformer for Image Change Captioning." IEEE Transactions on Image Processing (2025).
>
> [2] Yang, An, et al. "Qwen3 technical report." arXiv preprint arXiv:2505.09388 (2025).

---

### Official Review · Reviewer_syPv · 2025-11-01

**Soundness:** 2
**Presentation:** 3
**Contribution:** 2
**Rating:** 4
**Confidence:** 5

**Summary:**

This paper proposes ProCap, a two-stage framework that reformulates change captioning from static image comparison to dynamic procedure modeling. Stage 1 uses a pre-trained Frame Interpolation model to generate intermediate frames, samples informative keyframes via confidence scoring, and trains a Procedure Encoder through caption-conditioned masked reconstruction with three losses (reconstruction, alignment, consistency). Stage 2 replaces explicit frames with learnable procedure queries for efficient inference, fine-tuning the encoder end-to-end with a text decoder. The method achieves competitive results on three datasets, claiming that modeling "how" changes occur improves understanding beyond just comparing "what" changed.

**Strengths:**

1. Novel Paradigm and Strong Motivation: The paper presents a fresh perspective by shifting from static image comparison to dynamic procedure modeling, addressing a limitation in existing methods that ignore temporal dynamics.

2. Two-Stage Design Decoupling Learning and Inference: The framework separates explicit procedure learning (Stage 1) from implicit inference (Stage 2). Using learnable queries instead of generating frames at test time is technically sound.

3. Comprehensive Experimental Validation: The paper provides thorough evaluation across three diverse datasets, extensive ablations, rich visualizations (Figures 6-10), and detailed analysis of individual components. The experiments demonstrate consistent improvements over non-LLM baselines and competitive performance against LLM-based methods while being more efficient.

**Weaknesses:**

1.	This paper models the change procedure to improve captioning, but (132-137) the mapping γ_T: [0,1] → I is inherently non-bijective with an exponentially large solution space. For any given (I_bef, I_aft) pair, infinitely many valid procedures exist, yet the method relies on a generated sequence from FI without justifying why this particular realization should be canonical or optimal. This makes procedure modeling more difficult than directly describing the change—a small model must navigate a continuous, high-dimensional function space rather than a discrete caption space. Additionally, the confidence-based sampling assumes frames at the "semantic midpoint" (Eq. 2) are most informative, but no experiments validate this hypothesis.
2.	Table 2's ablation shows that adding implicit procedure queries alone (row 2, k=1) yields almost no improvement over the baseline (+2.2). The gain only appears in row 3 after incorporating the full Stage 1 pre-training. This demonstrates that the learnable queries contribute negligibly. The actual performance boost comes from the expensive Stage 1 pre-training itself, not from the "procedure modeling" concept.
3.	Stage 1 trains the encoder to reconstruct sampled frames, yet Stage 2 replaces them with learnable queries. If the frames are good training targets, why not use them directly for captioning as a simpler end-to-end approach? Additionally, the multi-granularity masking rationale (lines 236-237) is vague, and the specific schemes/probabilities (0.1, 0.7, 0.1, 0.1) appear without ablation justification.
4.	The model design is relatively complex, consisting of two training stages, with the first stage further divided into three submodules. Compared to other non-LLM-based models, the overall performance improvement is not significant, and some metrics (M, R) are relatively low.

**Questions:**

Please see the weaknesses.

---

> ### Author Response · Authors · 2025-11-21
> **Response to Reviewer syPv (Part 1)**
>
> We thank the reviewer for the insightful comments.
>
> ---
> **W1-1:** The mapping $\gamma_T: [0, 1]\rightarrow I$ is non-bijective with an exponentially large solution space, and a sequence generated by FI is not justified to be canonical or optimal. An exponentially large solution space makes procedure modeling more difficult than directly describing the change.
>
> **A:** We agree that without additional constraints, there exist infinite procedures for the same image pair. To restrict the solution space, we adopt an off-the-shelf optical-flow-based frame interpolation method to synthesize change procedures, **where the optical flow serves as a strong constraint**: the intermediate frame is obtained by warping the before and after images according to the linearly interpolated optical flow, rather than being generated from scratch.
>
> To empirically demonstrate the necessity of these constraints, we compared our approach with one diffusion-based frame interpolation, which operates within a significantly less constrained solution space. As shown in the following Table, relaxing the constraints leads to noticeable performance degradation compared to the optical-flow-based approach. We attribute this drop to the stochastic nature of diffusion models. Unlike optical-flow-based methods that enforce strict pixel-wise correspondence, diffusion models inherently introduce unpredictable and uncontrollable visual variations in the intermediate frames. These unintended variations make procedure modeling more difficult, hindering effective model training. We have included this analysis and qualitative examples generated by the diffusion-based frame interpolation in the revised version.
>
> | FI models | B $\uparrow$ | M $\uparrow$ | R $\uparrow$ | C $\uparrow$ |
> | --- | --- | --- | --- | --- |
> | Ours (Diffusion-based [1]) | 9.9 | 15.3 | 41.3 | 37.8 |
> | Ours (Optical-flow-based [2]) | **11.7** | **15.9** | **43.2** | **40.6** |
>
> [1] Zhang, Guozhen, et al. "Motion-aware generative frame interpolation." arXiv preprint arXiv:2501.03699 (2025).
>
> [2] Lu, Liying, et al. "Video frame interpolation with transformer." Proceedings of the IEEE/CVF Conference on Computer Vision and Pattern Recognition. 2022.
>
> ---
> **W1-2:** The assumption "'semantic midpoints' are most informative" has no experimental validation.
>
> **A:** We thank the reviewer for pointing this out and apologize for the confusion caused by our initial presentation. We clarify that the “semantic midpoints” our confidence-based strategy selected are considered relatively less information-redundant for the change analysis than those near the initial and final states (i.e.,$I_\text{bef}$ and $I_\text{aft}$). We have accordingly revised lines 197–200 in our paper.
>
> In Figure 4 in Appendix J.1, we compared our confidence-based strategy with random sampling and visual-only sampling, while varying the number of pseudo-frames $l$ on the CLEVR-Change dataset. The results are summarized in the table below (on CIDEr $\uparrow$):
>
> | Sample Strategy                       | 3     | 7     | 15    | 31    |
> |---|---|---|---|---|
> | Random Sampling                      | 135.2 | 135.5 | 128.5 | 128.0 |
> | Visual-only Sampling                 | 135.2 | 134.6 | 129.5 | 129.0 |
> | **Confidence-based Sampling (Ours)** | **135.2** | **135.6** | **135.2** | **135.7** |
>
>
> It can be observed that **our confidence-based strategy maintains consistently comparable CIDEr scores** while facing more candidate frames, whereas random sampling and visual-only sampling both exhibit noticeably larger performance degradation. This indicates that the frames (around semantic midpoints) selected by our strategy bring relatively more benefit for change analysis.
>
> We acknowledge that defining the theoretically 'optimal' informative point remains an open challenge. We have added a discussion on this to the Future Work section in the refined version.

---

> ### Author Response · Authors · 2025-11-21
> **Response to Reviewer syPv (Part 2)**
>
> **W2:** Table 2's results show the negligible contribution of learnable queries. The actual performance boost seems to come from the expensive Stage 1 pre-training rather than the "procedure modeling" concept.
>
> **A:** We would like to clarify that the performance boost is not solely from Stage 1 pre-training; it comes from the **synergy** between Stage 1 and the learnable queries in Stage 2, which **together instantiate the "procedure modeling" mechanism**. In the Table below, we have improved our initial presentation of Table 2. Here, explicit procedure modeling denotes Stage 1 training, and implicit procedure captioning denotes Stage 2 training with the learnable queries included. The number of learnable query sets $k$ is set to 1.
>
> Compared to the baseline initialized randomly, applying the learnable queries directly (line 3) introduces random vectors of learnable queries, therefore lacking any temporal or procedural context. In this case, the model cannot effectively reason about the evolution from the “before” to the “after” image.
>
> In contrast, applying Stage 1 without the learnable queries (line 2) demonstrates that pre-training alone provides only limited gains, far smaller than the improvement observed when *both* pre-training and learnable queries are used together (line 4). Thus, Stage 1 pre-training is necessary but not sufficient; the learnable queries give the model a mechanism to *use* the procedural knowledge learned in Stage 1. We have updated Table 2 and the analysis in the refined version.
>
>
> | Explicit procedure modeling (Stage 1) | Implicit procedure captioning (Stage 2) | B $\uparrow$ | M $\uparrow$ | R $\uparrow$ | C $\uparrow$ |
> |:---:|:---:|---|---|---|---|
> |                             |                                | 47.2 | 35.8 | 68.6 | 108.4 |
> | $\checkmark$                |                                | 52.6 | 38.0 | 70.1 | 112.7 |
> |                             | $\checkmark$                   | 47.3 | 36.3 | 68.8 | 106.2 |
> | $\checkmark$                | $\checkmark$                   | 56.5 | 41.9 | 75.5 | 128.5 |
>
> ---
> **W3-1:** Why not use the reconstructed sampled frames directly for captioning as a simpler end-to-end approach?
>
> **A:** While the reconstructed frames from Stage 1 are useful as training targets, using them directly for captioning is *not* preferable for two main reasons:
> 1. **Limited informativeness of synthesized frames.** The FI model in Stage 1 can reproduce change-procedure dynamics, but it may also generate trivial or redundant frames. Passing such imperfect frames to the captioner leads to notable performance degradation, as evidenced in Table 10 (Appendix K, +7.1 on CIDEr). In contrast, implicit queries retain only the salient temporal cues without propagating low-level redundancies.
> 2. **Computational inefficiency.** Using synthesized frames requires the image encoder to process more frame-level inputs, substantially increasing computational cost. As shown in Table 10, this explicit procedure-captioning pipeline is significantly slower than our implicit strategy (−278.01 TPS).
>
> Furthermore, in Stage 2, **the learnable queries are initialized from the Stage 1 masked embeddings**, allowing them to retain the learned procedure dynamics while being optimized for captioning. This yields better change modeling than directly consuming imperfect frames.
> For these reasons, the implicit procedure representation provides a more efficient and more effective alternative to using reconstructed frames directly.

---

> ### Author Response · Authors · 2025-11-21
> **Response to Reviewer syPv (Part 3)**
>
> **W4:** The model design is relatively complex and achieves no significant improvement compared to other non-LLM-based methods.
>
> **A:** We clarify that our model is **not more complex at inference time** than existing non-LLM-based methods, and the design is fully justified by clear **efficiency and effectiveness** advantages. Although the training process contains two stages, only the lightweight encoder-decoder from Stage 2 is used for inference. The components in Stage 1 (FI model, text embedding layer, image tokenizer) are auxiliary for training only and **are completely removed during inference**, leaving a structure directly comparable to existing non-LLM-based methods.
>
> To address the concern about the model’s complexity, we further performed a direct comparison on efficiency. As shown in the table below, **ProCap achieves both the fastest inference speed and the highest CIDEr score** compared to the strongest non-LLM-based methods (MCT-CCDiff [1]). ProCap is **22× faster** than MCT-CCDiff, while also outperforming both in CIDEr. This demonstrates that our architecture is not only not overly complex but in fact substantially more efficient at inference.
>
> | Method           | Speed $\downarrow$ (s/caption, used in MCT-CCDiff) | CIDER $\uparrow$ |
> |---|:---:|:---:|
> | MCT-CCDiff [1]   | 0.91 | 131.7 |
> | ProCap (Ours)    | 0.04 | 135.6 |
>
>
> In summary, the two-stage design adds no inference-time cost and ultimately yields a more efficient and more capable encoder-decoder than prior non-LLM-based SOTA methods.
>
> [1] Hu, Jinhong, et al. "MCT-CCDiff: Context-aware Contrastive Diffusion model with Mediator-bridging Cross-modal Transformer for Image Change Captioning." IEEE Transactions on Image Processing (2025).

---

> ### Author Response · Authors · 2025-11-25
> **Response to Reviewer syPv (Part 4)**
>
> **W3-2:** The multi-granularity masking rationale is vague and lacks ablation studies for probability settings of each masking strategy.
>
> **A:** We thank the reviewer for pointing this out. Our rationale for the multi-granularity masking design is **empirically demonstrated by ablation** (Table 7 in Appendix J.3, Spot-the-Diff dataset), and the chosen probabilities (0.1 / 0.7 / 0.1 / 0.1 for EM/RPM/IBM/OBM, representing entire masking, random patch masking, in-block masking, and out-of-block masking, respectively) are **not ad-hoc and can achieve a relatively better performance**. We have included an ablation study on the Spot-the-Diff dataset by varying the probability of each masking strategy, and we report the results in the table below.
>
> | EM  | RPM | IBM | OBM | B $\uparrow$  | M $\uparrow$  | R $\uparrow$   | C $\uparrow$   |
> |:-----:|:-----:|:-----:|:-----:|:-----:|:-----:|:------:|:------:|
> | 0.25 | 0.25 | 0.25 | 0.25 | 8.4 | 13.4 | 28.1 | 30.2 |
> | | | | | | | | |
> | 0.1 | 0.3 | 0.3 | 0.3 | 8.1 | 12.2 | 26.6 | 31.7 |
> | 0.4 | 0.2 | 0.2 | 0.2 | 7.9 | 11.7 | 28.0 | 28.9 |
> | 0.7 | 0.1 | 0.1 | 0.1 | 7.5 | 10.6 | 23.7 | 26.2 |
> | | | | | | | | |
> | 0.3 | 0.1 | 0.3 | 0.3 | 9.0 | 13.6 | 28.7 | 30.8 |
> | 0.2 | 0.4 | 0.2 | 0.2 | 8.3 | 11.3 | 28.1 | 35.7 |
> | 0.1 | 0.7 | 0.1 | 0.1 | **11.0** | **13.6** | **33.7** | **42.7** |
> | 0.0 | 1.0 | 0.0 | 0.0 | 8.5 | 12.1 | 28.2 | 31.5 |
> | | | | | | | | |
> | 0.3 | 0.3 | 0.1 | 0.3 | 8.3 | 12.1 | 28.1 | 31.6 |
> | 0.2 | 0.2 | 0.4 | 0.2 | 7.7 | 12.0 | 27.4 | 30.9 |
> | 0.1 | 0.1 | 0.7 | 0.1 | 8.1 | 11.9 | 27.3 | 29.0 |
> | | | | | | | | |
> | 0.3 | 0.3 | 0.3 | 0.1 | 8.1 | 12.8 | 31.9 | 33.1 |
> | 0.2 | 0.2 | 0.2 | 0.4 | 8.1 | 11.9 | 27.3 | 30.2 |
> | 0.1 | 0.1 | 0.1 | 0.7 | 7.9 | 12.5 | 27.4 | 30.7 |
>
>
> In combination with the original masking-strategy ablation (Table 7 in Appendix J.3, Spot-the-Diff dataset), the findings consistently show that:
> 1. **Stronger learning of distributed visual representations leads to better performance.** Random patch masking plays a central role in ensuring broad and dense visual coverage, and hence receives the highest probability.
> 2. **The entire masking, in-block masking, and out-of-block masking, are important** for capturing global context and localized structural information. However, dominating any of these structured strategies results in **losing too many fine-grained visual cues**, which **harms detailed feature learning** and negatively impacts change analysis, as can be found in the results that increasing the probability of any three masking strategies will lead to a performance degradation.
> 3. The four masking strategies function **synergistically** to support both coarse- and fine-grained representation learning, whereas using random patch masking alone provides only limited gains.
> 4. The chosen probability ratio (0.1 / 0.7 / 0.1 / 0.1) is therefore not ad-hoc; it is derived from the ablation as the configuration that: (1) preserves essential coarse and fine-grained cues, and (2) maximizes coverage of distributed representations.
>
> This updated explanation should clarify the rationale behind the multi-granularity masking design and directly justify the probability scheme. We have contained the analysis in the revised version.

---

### Author Response · Authors · 2025-12-03

Dear Program Chairs, Senior Area Chairs, Area Chairs and Reviewers,

We thank the reviewers for their valuable comments and constructive feedback. We are encouraged that all reviewers recognized the value and novelty of our work. Specifically, we would like to highlight the following strengths acknowledged by the reviewers:

1. **Novel Problem Formulation** `(syPv, REM5, weBZ, and hLpn)`. Our paradigm shift from static comparison to dynamic procedure modeling was highly appreciated.
2. **Soundness of Architecture** `(syPv, REM5, weBZ, and hLpn)`. The efficacy of our two-stage framework design was confirmed by the reviewers.
3. **Comprehensive Experimental Validation** `(syPv and hLpn)`. Our extensive evaluations were acknowledged for demonstrating the robustness and consistent superiority of our method.

During the rebuttal phase, we address concerns through supplementary experiments, analyses, and clarifications. The summary of major concerns, our response, and paper updates are listed below:

1. **Motivation & Feasibility** `(syPv, hLpn)`: We have addressed the concerns regarding the infinite solution space in continuous change procedures by providing in-depth discussions and additional comparative experiments in Appendix K.2.
2. **Spot-the-Diff Performance** `(syPv, REM5, weBZ, and hLpn)`: We have analyzed the performance gap compared to MCT-CCDiff [1] on the Spot-the-Diff dataset, supported by comprehensive comparisons under aligned settings in Appendix J.
3. **Supplementary Experiments** `(syPv, weBZ, and hLpn)`: We have enriched the manuscript with failure case studies, analyses of significant viewpoint shifts, and extensive ablation studies (particularly on Spot-the-Diff). All revisions are highlighted in blue.
4. **Future Work Discussion** `(REM5 and weBZ)`: We have expanded the discussion on future directions in Appendix N, specifically focusing on LLM-backbone integration and the handling of extreme scene differences.

We are particularly grateful to reviewer hLpn for the active engagement and for raising the score from 6 to 8. We thank all the reviewers, ACs, SACs, and PCs for their time and dedication to the review process.

Best regards,

Authors of Paper Submission 7411

---
[1] Hu, Jinhong, et al. "MCT-CCDiff: Context-aware Contrastive Diffusion model with Mediator-bridging Cross-modal Transformer for Image Change Captioning." IEEE Transactions on Image Processing (2025).

---

### Meta-Review · Area_Chair_AKuj · 2025-12-26

**Summary:**

syPv: (1) Concern on non-bijective mapping making procedure modeling more difficult than. directly describing the image. (2) The performance boost comes from Stage 1 pre-training, not from "procedure modeling" according to the ablation. (3) Disconnect between Stage 1 and Stage 2. (4) Model is complex, performance improvement is limited.

REM5: (1) performance is not consistent across three datasets. (2) ProCap used non-LLM-based backbone. Unclear whether it still perform when using LLM-based backbone.

weBZ: (1) does not show clear performance advantage on Spot-the-Diff over other SOTA methods. (2) consistency loss does not yield a substantial performance gain (table 4)

hLpn: (1) performance is worse on Spot-the-Diff. (2) Should present some failure cases.

**Reviewer Concerns:**

syPv: this reviewer's concerns are mostly addressed by the rebuttal.

REM5: this reviewer's concerns are mostly addressed.

weBZ: the concerns are mostly addressed.

hLpn: the concerns are mostly addressed. The reviewer explicitly mentioned raising the score.

**Reviewer Scores:**

Reviewer hLpn explicitly mention raising the score.

For other three reviewers, the scores likely will increase as well.

---

### Decision · Program_Chairs · 2026-01-26

Accept (Poster)